# LABEL-AGNOSTIC FORGETTING: A SUPERVISION-FREE UNLEARNING IN DEEP MODELS

**Shaofei Shen**[1]     **Chenhao Zhang**[1]     **Yawen Zhao**[1]     **Alina Bialkowski**[1]
**Weitong Chen**[2]     **Miao Xu**[1]*
[1]University of Queensland     [2]University of Adelaide
{shaofei.shen,chenhao.zhang,yawen.zhao,alina.bialkowski}@uq.edu.au
t.chen@adelaide.edu.au, miao.xu@uq.edu.au

## ABSTRACT

Machine unlearning aims to remove information derived from forgotten data while preserving that of the remaining dataset in a well-trained model. With the increasing emphasis on data privacy, several approaches to machine unlearning have emerged. However, these methods typically rely on complete supervision throughout the unlearning process. Unfortunately, obtaining such supervision, whether for the forgetting or remaining data, can be impractical due to the substantial cost associated with annotating real-world datasets. This challenge prompts us to propose a supervision-free unlearning approach that operates without the need for labels during the unlearning process. Specifically, we introduce a variational approach to approximate the distribution of representations for the remaining data. Leveraging this approximation, we adapt the original model to eliminate information from the forgotten data at the representation level. To further address the issue of lacking supervision information, which hinders alignment with ground truth, we introduce a contrastive loss to facilitate the matching of representations between the remaining data and those of the original model, thus preserving predictive performance. Experimental results across various unlearning tasks demonstrate the effectiveness of our proposed method, Label-Agnostic Forgetting (LAF) without using any labels, which achieves comparable performance to state-of-the-art methods that rely on full supervision information. Furthermore, our approach excels in semi-supervised scenarios, leveraging limited supervision information to outperform fully supervised baselines. This work not only showcases the viability of supervision-free unlearning in deep models but also opens up a new possibility for future research in unlearning at the representation level.

## 1 INTRODUCTION

Currently, machine unlearning has attracted increasing attention due to rising concerns about data privacy issues (Xu et al., 2024; Nguyen et al., 2022; Bourtoule et al., 2021). To protect the privacy and interest of the owner of sensitive data, legislators in many regions have introduced laws like GDPR (Voigt & Von dem Bussche, 2017), and CCPA (de la Torre, 2018), which demand the deletion of sensitive information from the well-trained models.

The objective of machine unlearning is to remove information associated with *forgetting data* from the original model while preserving the knowledge contained in the *remaining data* (Bourtoule et al., 2021). A direct and intuitive strategy to achieve this is to retrain a new model from scratch utilizing only the remaining dataset. However, this method can be both time-consuming and computationally demanding (Zhang et al., 2022; Di et al., 2022). Without doing retraining, existing works on machine unlearning can be divided into two types. The first type is *exact unlearning* (Bourtoule et al., 2021; Kim & Woo, 2022). This approach necessitates that the unlearned model attains exactly the same performance as the retrained model, with respect to both model parameters and prediction accuracy (Bourtoule et al., 2021). In deep learning models, the exact unlearning is usually achieved through a distributed retraining strategy (Bourtoule et al., 2021). The second type is termed as *approximate*

---

*Corresponding author

*unlearning*, which requires the unlearned model to get similar prediction performances to the retrained model on not only the remaining data but also the forgetting data (Thudi et al., 2022; Chundawat et al., 2023; Kurmanji et al., 2023). Current strategies for approximate unlearning encompass methods such as training teacher models to facilitate the removal of forgetting data (Chundawat et al., 2023; Kurmanji et al., 2023), or retracing the alterations of parameters occurring in the training of forgetting data to reverse its effect of training (Thudi et al., 2022). Although both methods have achieved notable performance, it is worth noting that both types of work rely on the annotated remaining and forgetting data to guide the removal of undesired information and preservation of other necessary information. This line of works can be regarded as *supervised unlearning*.

While both types of work have demonstrated commendable performance, it is imperative to acknowledge the prevalent reality: in the real world, a significant portion of data remains unannotated, leading to a substantial number of machine learning models being trained on weakly labelled data (Nodet et al., 2020). This situation is exemplified in semi-supervised learning, which capitalizes on a vast pool of unlabelled data alongside a smaller set of annotated data for training purposes (Yang et al., 2023). Moreover, in the pursuit of privacy protection, even when training data is fully labelled, these labels may not be accessible during the unlearning phase. Previous unlearning works have necessitated the use of label information as optimization targets, either for purging information related to forgotten data or for retaining knowledge about the remaining data (Thudi et al., 2022; Chundawat et al., 2023; Chen et al., 2023; Kurmanji et al., 2023). Consequently, studies focused on supervised unlearning are limited in their ability to execute the unlearning task or preserve prediction performance in the absence of sufficient supervision information. This underscores the critical need for an unlearning algorithm that operates without relying on supervision information during the unlearning process, which we term as *label-agnostic unlearning*.

Therefore, we propose a framework named Label-Agnostic Forgetting (LAF)[1] for the label-agnostic unlearning, which can accomplish the unlearning task without the supervision information. Specifically, we alleviate the dependency of the unlearning process on supervision information by adjusting the representation distributions of the representation extractor, rather than changing the classifiers. We utilize a variational inference approach (Kingma & Welling, 2014) to estimate the representation distribution of the remaining data and then design an extractor unlearning loss to adjust the representation extractor, aiming to eliminate the information associated with the forgetting data at the representation level. To retain the prediction performances of the model after shifting representations, we devise a contrastive loss to align the remaining data representations of the adjusted model with those of the original model. Furthermore, if the supervision information is available, we can further adjust the classifier to fit the output distributions with the ground truths.

The contributions of this paper can be summarised as follows:

- Addressing the research gap in label-agnostic unlearning, we introduce and propose a framework named LAF, which is capable of accomplishing unlearning tasks and retaining high predictive performance post-learning, all without the need for supervision information. The proposed LAF can work effectively for mainstream unlearning problems.
- We incorporate the variational inference and contrastive learning approaches and propose two novel loss functions for extractor unlearning and representation alignment.
- Through empirical evaluations across multiple datasets and models, we demonstrate that LAF is comparable with full supervised unlearning methods. In addition, when limited supervision information is available, the LAF can outperform other state-of-the-art works.

## 2 PRELIMINARY

Let $D$ denote the training data which can be either fully supervised or semi-supervised, and $g_D$ represents a deep model trained on $D$, which maps an instance $x \in \mathcal{X}$ to a label $y \in \mathcal{Y}$. The unlearning on the deep model $g_D$ aims to remove the knowledge related to forgetting data $D_f \subset D$, and preserve the knowledge learned from the remaining data $D_r = D - D_f$. Assuming that the training data $D$, remaining data $D_r$ and forgetting data $D_f$ are sampled from the distributions $\mathcal{P}$, $\mathcal{P}_r$ and $\mathcal{P}_f$ respectively, the machine unlearning algorithm $U$ should yield a deep model $g_U = U(g_D, D_r, D_f)$ which approximates the performance of $g_{D_r}$ trained on $D_r$ only. That is, $g_U(x) = g_{D_r}(x)$ no matter $x$ is sampled from $\mathcal{P}_r$ or $\mathcal{P}_f$. From an intuitive sense, $g_U(x)$ should be similar to $g_D(x)$ for $x \sim \mathcal{P}_r$,

---

[1]https://github.com/ShaofeiShen768/LAF

but (significantly) different for $x \sim \mathcal{P}_f$, such that the forgetting effect can be achieved without impacting performance on the remaining data.

As a deep model, $g$ can be viewed as the concatenation of two main components: a representation extractor $g_D^e$ and a downstream classifier $g_D^c$. In the case of label-agnostic unlearning, wherein labels may not be readily accessible or reliable during the unlearning phase, a potential approach to address this challenge is to adjust the representation extractor $g_D^e$ to eliminate the memory of forgotten data.

There are three main challenges when adjusting $g_D^e$. The first involves estimating the knowledge associated with forgetting data within the representation extractor $g_D^e$. Secondly, the process of removing the knowledge of the forgetting data from $g_D^e$ lacks a well-defined optimization objective. Traditional objectives used in supervised unlearning scenarios may not apply, especially in cases where label information is either unavailable or unreliable in label-agnostic unlearning situations. Thirdly, modifying $g_D^e$ can also impact the representations of the remaining data, potentially causing a misalignment with the classifier $g_D^c$ and consequently leading to a decrease in predictive performance.

## 3 Methodology

In this section, we tackle the three aforementioned challenges by introducing the Label Agnostic Forgetting (LAF) method, which comprises two updates: the extractor unlearning and the representation alignment. Importantly, both of these updates operate without the need for supervision information. During the extractor unlearning stage, we estimate the distribution of representations for both the forgetting data and the remaining data, leveraging the original model's knowledge acquired from these distinct data groups. Subsequently, we introduce two objectives to facilitate unlearning, with another proposed extractor unlearning loss. Moving on to the representation alignment stage, we recognize that alterations in representation may impact the alignment between these representations and the classifiers. To address this, we propose a contrastive loss that aligns the representations post-unlearning with those pre-unlearning, preserving predictive performance in light of the absence of label information. Furthermore, we consider scenarios where limited supervision information is available. In such cases, we incorporate an additional supervised repair step to further enhance the unlearning performance.

The subsequent subsections will delve into the specifics of extractor unlearning, and representation alignment, and provide an overview of the complete LAF algorithm.

### 3.1 Extractor Unlearning

In the extractor unlearning stage, we first discuss the relationship between the data's distribution and post-unlearning extractor $g_U^e(\cdot)$. Assuming that for $x \sim \mathcal{P}_r$, $g_U^e(x)$ follows a distribution $Q(D_r)$ and for $x \sim \mathcal{P}_f$, $g_U^e(x)$ follows a distribution $Q(D_f)$, then one possible way to learn the optimal $\theta^*$ for $g_U^e$ parameterized on $\theta$ can be two-objective, that is,

$$\min_\theta \Delta(Q(D_r), \mathcal{P}_r), \text{ where } x \sim \mathcal{P}_r, g_U^e(x) \sim Q(D_r), \text{ and simultaneously} \tag{1}$$

$$\max_\theta \Delta(Q(D_f), \mathcal{P}_f), \text{ where } x \sim \mathcal{P}_f, g_U^e(x) \sim Q(D_f). \tag{2}$$

In the above two equations, $\Delta(\cdot, \cdot)$ represents the discrepancy between two distributions. Eq. 1 and Eq. 2 describe the intuition that the unlearning extractor should attain the distribution of the remaining data but dissolve the distribution of forgetting data. We preserve the knowledge of the remaining data through Eq. 1, and treat the forgetting data as irrelevant data via Eq. 2. In this way, the forgetting data will be predicted based on the preserved knowledge instead of random guessing. We could use multi-objective solvers (Coello et al., 2002) for the optimization problem Eqs. 1 and 2. In this paper, considering the benefit of end-to-end models, we merge these two objectives into the following one for learning the optimal $\theta^*$

$$\theta^* = \arg\min_\theta \Delta(Q(D_r), \mathcal{P}_r) - \Delta(Q(D_f), \mathcal{P}_f). \tag{3}$$

Since the size of $D_f$ is limited, training new models on $D_f$ becomes challenging. In addition, training new models on $D_f$ can also be inefficient. We cannot directly have $\mathcal{P}_r$ and $\mathcal{P}_f$ and thus we first estimate them through the representation extractor of the $g_D$, i.e., $g_D^e$. This estimation is based on

the assumption that $g_D$ can convey sufficient information on the data that it is trained on. However, $g_D^e$ is a well-trained representation extractor, which cannot be used directly to approximate the data distribution. Moreover, there could be a discrepancy between the distribution of the forgetting data and the remaining data. Thus we need a model to catch the difference accurately.

With such a goal to mimic the distribution and capture the difference between $D_r$ and $D_f$, we train two VAEs (Kingma & Welling, 2014) to approximate the distribution of representations of $D_r$ and $D_f$. Specifically, we first train VAE $h$ for the remaining data $D_r$. Note that instead of capturing the information with $D_r$, $h$ captures the information of all training data $D$. This enables direct training of the VAE without the need to specify forgetting data. Such efficiency gain is crucial since computational resources and time are significant considerations. In addition, the number of forgetting data is always far less than the number of training data; thus $h$ can sufficiently represent the distribution $\mathcal{P}_r$. Therefore,, we use the entire dataset $D$ for training VAE $h$, leading to the following optimization function:

$$\arg\min_h \mathbb{E}_{z\sim\mathcal{N}(\mu_h,\sigma_h^2)} \log P_h(g_D^e(x_r)|z) + KL(\mathcal{N}(\mu_h,\sigma_h^2)||\mathcal{N}(0,\mathcal{I})), \qquad (4)$$

where $h$ outputs a representation for any $x_r \sim \mathcal{P}$, $\mathcal{N}(0,\mathcal{I})$ is the standard Gaussian distribution, $\mathcal{N}(\mu_h,\sigma_h^2)$ is the Gaussian distribution parameterized on $\mu_h$ and $\sigma_h$, and $z$ is the reparameterized sample from the Gaussian distribution. Here $\mu_h$ and $\sigma_h$ are the mean and standard deviation estimated by $h$ on its encoding layer for $g_D^e(x), x \in D$.

In parallel, another VAE $h_f$ is trained specifically for the representations of forgetting data $D_f$ by

$$\arg\min_{h_f} \mathbb{E}_{z\sim\mathcal{N}(\mu_{h_f},\sigma_{h_f}^2)} \log P_{h_f}(g_D^e(x_f)|z) + KL(\mathcal{N}(\mu_{h_f},\sigma_{h_f}^2)||\mathcal{N}(0,\mathcal{I})), \qquad (5)$$

where $\mu_{h_f}$ and $\sigma_{h_f}$ are the mean and standard deviation estimated by $h_f$ on its encoding layer for $g_D^e(x_f), x_f \in D_f$.

After we learned $h$, which captures the distribution of representations extracted by $g_D^e(x)$, we still need to have $Q(D_r)$, which is the distribution of representations extracted by $g_U^e(x)$ on the remaining data. As in the $\mathcal{P}$ case, the representation extractor itself cannot be used as distributions, and we still need another VAE to express that. Since we cannot learn a new VAE for the unknown $g_U^e$, we propose to use the VAE learned in Eq. 4 to describe the distribution on $D_r$. In this way, by fixing the VAE $h$, the objective in Eq. 1 to learn the $g_U^e$ can be

$$\arg\min_\theta \mathbb{E}_{z\sim\mathcal{N}(\tilde{\mu}_h,\tilde{\sigma}_h^2)} \log P_h(g_U^e(x_r)|z) + KL(\mathcal{N}(\tilde{\mu}_h,\tilde{\sigma}_h^2)||\mathcal{N}(0,\mathcal{I})), \qquad (6)$$

where $\tilde{\mu}_h$ and $\tilde{\sigma}_h$ are the mean and standard deviation estimated by $h$ on its encoding layer for $g_U^e(x_r), x_r \in D_r$. Another objective corresponding to Eq. 2 can be optimized by

$$\arg\max_\theta \mathbb{E}_{z\sim\mathcal{N}(\tilde{\mu}_{h_f},\tilde{\sigma}_{h_f}^2)} \log P_{h_f}(g_U^e(x_f)|z) + KL(\mathcal{N}(\tilde{\mu}_{h_f},\tilde{\sigma}_{h_f}^2)||\mathcal{N}(0,\mathcal{I})), \qquad (7)$$

where $\tilde{\mu}_{h_f}$ and $\tilde{\sigma}_{h_f}$ are the mean and standard deviation estimated by $h_f$ on its encoding layer for $g_U^e(x_f), x_f \in D_f$.

We can then merge Eqs 6 and 7 in the same way as Eq. 3 into one overall objective

$$\theta^* = \arg\min_\theta (\mathbb{E}_{z\sim\mathcal{N}(\tilde{\mu}_h,\tilde{\sigma}_h^2)} \log P_h(g_U^e(x_r)|z) - \mathbb{E}_{z\sim\mathcal{N}(\tilde{\mu}_{h_f},\tilde{\sigma}_{h_f}^2)} \log P_{h_f}(g_U^e(x_f)|z) +$$
$$KL(\mathcal{N}(\tilde{\mu}_h,\tilde{\sigma}_h^2)||\mathcal{N}(0,\mathcal{I})) - KL(\mathcal{N}(\tilde{\mu}_{h_f},\tilde{\sigma}_{h_f}^2))||\mathcal{N}(0,\mathcal{I}))), \qquad (8)$$

The second part, $KL(\mathcal{N}(\tilde{\mu}_h,\tilde{\sigma}_h^2)||\mathcal{N}(0,\mathcal{I})) - KL(\mathcal{N}(\tilde{\mu}_{h_f},\tilde{\sigma}_{h_f}^2))||\mathcal{N}(0,\mathcal{I}))$, typically act as penalty terms, enforcing a regularization effect. By eliminating these terms, we aim to reduce the constraints on the model, thereby allowing a more flexible adjustment of the distribution during the unlearning process. Hence, we simplify the objective in Eq. 8 by removing the second part. Noticed that we only drop the two KL divergence terms for unlearning as follows and we use the complete Eq. 4 and Eq. 5 for the VAE training:

$$\theta^* \approx \arg\min_\theta (\mathbb{E}_{z\sim\mathcal{N}(\tilde{\mu}_h,\tilde{\sigma}_h^2)} \log P_h(g_U^e(x_r)|z) - \mathbb{E}_{z\sim\mathcal{N}(\tilde{\mu}_{h_f},\tilde{\sigma}_{h_f}^2)} \log P_{h_f}(g_U^e(x_f)|z)). \qquad (9)$$

In implementations, the log-likelihood is usually optimized by the L2 loss as the reconstruction loss of the input and output of the VAE. Furthermore, in the unlearning stage, the negative $L_2$ loss

can easily diverge and then lead to the breakdown of the whole deep model, which is known as *catastrophic unlearning* (Nguyen et al., 2020). Therefore, to avoid catastrophic unlearning, we propose a normalized form of $L_2$ loss and optimize $\theta^*$ by the extractor unlearning loss $L_{UE}$:

$$L_{UE} = \sum_{x \in X_r} \frac{\|g_U^e(x) - h(g_U^e(x))\|_2^2}{\|g_U^e(x) - h(g_U^e(x))\|_2^2 + 1} - \sum_{x \in X_f} \frac{\|g_U^e(x) - h_f(g_U^e(x))\|_2^2}{\|g_U^e(x) - h_f(g_U^e(x))\|_2^2 + 1}, \qquad (10)$$

where $X_r$ and $X_f$ denote the inputs to the model of remaining data and forgetting data.

## 3.2 REPRESENTATION ALIGNMENT

On the one hand, the extractor unlearning stage introduces two approximate estimations of the representation distribution $Q(D_r)$ and $Q(D_f)$. $g_U^e(x)$ necessitates further adjustment to mitigate the influences induced by this approximate estimation. On the other hand, after the extractor unlearning, the representation space of model $g_U^e$ can not be aligned with the original classifier layers $g_D^c$. The adjusted model will suffer a performance drop in the predictions. Furthermore, due to the lack of supervision information $y$, the original classifier $g_D^c$ cannot be adjusted to align with the representation space after the extractor unlearning to retain the prediction capability of the whole model. To maintain the prediction performances of updated models, one possible approach is to shift the representation space after the extractor unlearning to align with the original representation space on the remaining data. In this case, the adjusted model after the extractor unlearning can utilize the supervision knowledge of the original model without adjusting $g_D^c$ using any label information. Therefore, we propose the representation alignment loss function and optimize $g_U^e$ by minimizing the proposed loss:

$$L_{RA} = \sum_{x \in X_r} \log\left(\frac{\exp(simloss(g_U^e(x), g_D^e(x)))}{\sum_{\hat{x} \in X_f} \exp(simloss(g_U^e(\hat{x}), g_D^e(\hat{x}))/\tau)}\right), \qquad (11)$$

where $\tau$ is a hyperparameter. As for the similarity loss function $simloss(\cdot, \cdot)$ in Eq. 11, we use the cosine similarity loss for implementations because of its normalization characteristics and huge success on the high dimensional representation learning (Chen et al., 2020). Different from the classical contrastive loss, the representation alignment loss compares the similarity of the representations of the models before and after the unlearning algorithm on the same data point. The representation alignment loss reduces the distance between the representations of the two models on the remaining data point and increases the dissimilarity between the representations of the two models on the forgetting data. In the implementation, we optimized the extractor unlearning loss $L_{UE}$ and representation alignment loss $L_{RA}$ alternately because the magnitudes of the two losses have large differences in different datasets.

For the training data with extra annotations, we add an additional supervised repairing stage to retain higher performances after the LAF if the labels of the part of the remaining data are available. In the implementation, we sample the same number of remaining data as the number of forgetting data $D_f$.

---

**Algorithm 1** Supervision-free Unlearning Algorithm: LAF

---

**Input:**
    The training data, $D$, consisting of the remaining data $D_r$ and forgetting data $D_f$;
    Two initialized VAEs, $h$ and $h_f$;
    The original model $g_D$ with the representation extractor $g_D^e$;
    Epochs for the unlearning, $Epoch_r$.
**Output:**
    The updated model that removes the knowledge of the forgetting data, $f_{\theta^*}$.
1: Train $h$ and $h_f$ on $D$ and $D_f$ by Eq. 4 and Eq. 5;
2: Fix $h$ and $h_f$ and set $g_U = g_D$;
3: **for** $e_r$ in $range(Epoch_r)$:
4:     Sample from $D_r$ and get samples as the same size as $D_f$;
5:     Update $g_U^e$ by Eq. 10;
6:     Update $g_U^e$ by Eq. 11;
7: **if** labels of partial remaining data are available:
8:     Repair $g_U$ by supervised data for one epoch;
9: **return** $g_U$.

---

### 3.3 OVERALL ALGORITHM AND IMPLEMENTATION

We provide the pseudo-code of LAF in Algorithm 1. The LAF takes the inputs of training data $D$ and inputs of forgetting data $D_f$ to get the embeddings through the extractor $g_U^e$. Then these embeddings work as the inputs to train two VAEs using optimization functions (Eq. 4 and Eq. 5). The two VAEs are expected to learn the distribution of original training data and forgetting data. We then remove the knowledge of the forgetting data while preserving the knowledge of the remaining data via the extractor unlearning loss in Eq. 10. Subsequently, to maintain the performance of the post-unlearning model, we align the representation space of the remaining data with the original representation space via the representation alignment loss in Eq. 11. Furthermore, when the supervision information of the remaining data is available, the post-unlearning model can be further repaired.

## 4 EXPERIMENTS

In this section, we conduct experiments to answer three research questions to evaluate LAF:

- **RQ1**: How do the proposed LAF perform on the data removal, class removal, and noisy label removal tasks, as compared with the state-of-the-art unlearning methods?
- **RQ2**: Is the representation space after the implementation of LAF consistent with the space of retrained models?
- **RQ3**: How do $L_{UE}$ and $L_{RA}$ affect the performance of LAF?

### 4.1 SETTINGS

**Datasets and Models.** To validate the effectiveness of LAF, we conduct experiments on four datasets: **DIGITS** (MNIST) (LeCun, 1998), **FASHION** (Fashion MNIST)(Xiao et al., 2017), **CIFAR10** (Krizhevsky et al., 2009) and **SVHN** (Netzer et al., 2011). For the two MNIST datasets, we use a convolutional neural network (**CNN**) with two convolutional layers while for the two CIFAR datasets, we choose an **18-layer ResNet** backbone(He et al., 2016).

**Baselines.** Considering that the label-agnostic unlearning on deep models is still a research gap, we compare the performance of **LAF** (label-agnostic) and **LAF+R** (with supervised data for repairing) with seven fully supervised unlearning baselines including **Retrain** which are the golden standards from the retraining models, and six state-of-the-art unlearning works on deep models. The six baselines include four approximate unlearning works that have been published in the past year and one exact unlearning method: **NegGrad**, **Boundary** (Chen et al., 2023), **SISA** (Bourtoule et al., 2021), **Unroll** (Thudi et al., 2022), **T-S** (Chundawat et al., 2023), and **SCRUB** (Kurmanji et al., 2023). The detailed descriptions of these baselines and the implementation details are provided in Appendix 3. All the experiments on these baselines are conducted for five rounds of different random seeds.

**Evaluation Setting.** To validate the efficacy of the LAF, we establish experiments under three scenarios: (1) data removal, wherein 40% of training data labelled from 5 to 9 are randomly selected for removal; (2) class removal, where data from class 0 are designated as forgetting data for removal; (3) noisy label removal, in which 60% of training data labelled from 0 to 4 are randomly annotated as wrong labels and regarded as forgetting data for removal. For evaluations, we assess the performance of LAF in comparison to baseline methods using four metrics: **Train$_r$**, representing the prediction accuracy of the post-unlearning model on the remaining data; **Train$_f$**, denoting the prediction accuracy of the post-unlearning model on the forgetting data; **Test**, indicating the prediction accuracy of the post-unlearning model on the test data, which is further divided into **Test$_r$** and **Test$_f$** in the class removal task, representing test accuracy on the remaining and forgetting classes respectively; and **ASR**, which denotes the attack success rate of the membership inference attack (Shokri et al., 2017; Chen et al., 2021). For all the above four metrics, the closer value on **Train$_f$** and **ASR** of the post-unlearning model to the retrained model indicates better knowledge removal performance while the closer value on **Test** to the retrained model indicates better knowledge preservation performance.

### 4.2 UNLEARNING PERFORMANCES

Table 1, 2, and 3 showcase the experiment results of the three distinct tasks: data removal, class removal, and noisy label removal. Upon comprehensive analysis of the experimental outcomes, it is

Table 1: Comparison results with other state-of-the-art methods in data removal (avg%±std%). The **bold** record indicates the best result and the underlined record indicates the second best result. The following tables use the same notations as this table.

| Method | Data | Train_r | Train_f | Test | ASR | Data | Train_r | Train_f | Test | ASR |
|---|---|---|---|---|---|---|---|---|---|---|
| Retrain | DIGITS | 99.56±0.05 | 98.84±0.10 | 99.04±0.10 | 49.80±0.53 | FASHION | 96.43±0.35 | 92.15±0.41 | 90.23±0.22 | 47.32±0.76 |
| NegGrad | | 99.18±0.28 | **98.86±0.41** | 98.62±0.29 | 50.24±0.27 | | 93.28±0.29 | 88.93±0.79 | 89.18±0.24 | 46.11±0.66 |
| Boundary | | 97.65±1.02 | 95.36±2.50 | 96.63±1.35 | 46.83±2.09 | | 56.28±4.69 | 46.58±4.04 | 53.00±3.66 | 48.03±1.41 |
| SISA | | 99.06±0.12 | 98.60±0.07 | 98.92±0.02 | 33.78±0.01 | | 91.98±0.19 | 90.76±0.07 | 89.92±0.24 | 33.33±0.02 |
| Unrolling | | **99.63±0.15** | 99.34±0.33 | **99.08±0.18** | 46.50±0.60 | | 89.83±0.30 | 83.88±0.65 | 81.21±0.34 | 47.69±0.50 |
| T-S | | 94.01±0.77 | 93.09±2.73 | 93.72±1.03 | 47.82±0.64 | | 82.96±1.14 | 86.77±2.13 | 82.46±1.24 | 45.90±1.30 |
| SCRUB | | 99.28±0.04 | 99.03±0.12 | 98.95±0.08 | 46.68±0.80 | | 90.88±0.09 | 88.62±0.28 | 88.75±0.11 | 45.23±0.94 |
| **LAF+R** | | 99.47±0.14 | 99.35±0.65 | 98.89±0.10 | 49.42±0.51 | | **94.18±0.30** | 95.00±1.62 | **90.51±0.28** | 47.39±0.23 |
| **LAF** | | 98.03±0.68 | 97.29±1.43 | 97.30±0.78 | 47.92±0.84 | | 91.54±2.67 | **90.91±7.00** | 87.53±3.26 | 46.89±0.88 |
| Retrain | CIFAR10 | 84.03±0.20 | 78.05±1.34 | 87.20±0.65 | 57.48±0.88 | SVHN | 83.88±0.23 | 75.16±0.76 | 93.41±0.40 | 58.76±0.48 |
| NegGrad | | 79.08±0.55 | 70.50±2.94 | 83.51±0.97 | 56.53±0.34 | | 81.57±0.34 | 69.93±1.66 | 91.54±1.01 | 57.94±0.80 |
| Boundary | | 54.73±1.32 | 18.73±3.33 | 51.23±2.55 | 62.79±0.95 | | 64.85±2.06 | 28.62±1.89 | 73.07±1.96 | 89.17±3.29 |
| SISA | | 66.78±0.10 | 53.12±0.74 | 54.30±0.05 | 37.53±0.02 | | 82.48±0.17 | 67.79±0.34 | 82.57±0.83 | 50.19±0.38 |
| Unrolling | | 57.82±1.66 | 30.91±2.86 | 61.31±1.51 | 56.97±1.27 | | 70.98±1.87 | 47.68±2.72 | 83.27±0.48 | 55.39±0.98 |
| T-S | | 70.31±2.32 | 72.17±3.91 | 77.71±2.02 | 54.64±1.58 | | 78.36±0.13 | 73.50±0.62 | 90.60±0.61 | 55.77±1.42 |
| SCRUB | | 29.16±1.07 | 0.47±0.93 | 25.18±0.78 | 54.03±0.64 | | 22.32±0.04 | 0±0 | 19.59±0.07 | 65.26±1.24 |
| **LAF+R** | | **79.57±0.72** | **79.50±0.66** | **84.74±1.08** | 57.74±0.62 | | **83.37±0.41** | **76.08±0.76** | **93.56±0.51** | **58.03±0.28** |
| **LAF** | | 78.03±1.55 | 73.30±3.96 | 82.22±2.57 | **57.65±0.70** | | 81.63±0.49 | 76.11±1.49 | 92.32±0.58 | 57.85±0.89 |

Table 2: Comparison results with other state-of-the-art methods in class removal (avg%±std%)

| Method | Data | Test_r | Test_f | ASR | Data | Test_r | Test_f | ASR |
|---|---|---|---|---|---|---|---|---|
| Retrain | DIGITS | 98.81±0.15 | 0±0 | 26.49±1.41 | FASHION | 92.66±0.29 | 0±0 | 38.24±3.13 |
| NegGrad | | 98.86±0.39 | 79.76±38.91 | 39.71±3.99 | | 89.70±0.74 | 0.92±0.48 | **37.64±2.32** |
| Boundary | | 98.59±0.23 | 95.63±5.27 | 38.51±4.25 | | 86.04±1.41 | 1.68±0.69 | 39.06±2.57 |
| SISA | | **99.10±0.03** | **0±0** | 50.12±0.23 | | **92.14±0.07** | **0±0** | 50.00±0.02 |
| Unrolling | | 97.05±1.25 | 79.63±39.00 | 40.16±2.52 | | 88.72±0.86 | 0.40±0.24 | 40.61±1.87 |
| T-S | | 61.31±40.52 | 0.16±0.19 | 35.83±11.47 | | 91.84±0.31 | 21.16±7.24 | 24.82±0.61 |
| SCRUB | | 99.02±0.06 | 95.41±3.35 | 32.04±2.75 | | 91.40±0.24 | 0.42±0.36 | 33.84±0.60 |
| **LAF+R** | | 99.05±0.04 | 0.10±0.13 | **24.38±0.34** | | 91.95±0.29 | 0.08±0.08 | 35.00±2.16 |
| **LAF** | | 98.03±0.20 | 0.26±0.11 | 52.25±2.61 | | 89.73±0.37 | 2.43±1.46 | 31.35±0.71 |
| Retrain | CIFAR10 | 87.01±0.64 | 0±0 | 67.76±1.58 | SVHN | 94.07±0.67 | 0±0 | 59.33±1.31 |
| NegGrad | | 57.55±2.84 | **0±0** | 50.46±0.81 | | 76.92±1.23 | 6.44±9.12 | 52.70±3.62 |
| Boundary | | 83.33±1.36 | 1.00±0.66 | 61.22±3.37 | | 90.59±1.32 | 15.99±5.01 | 60.88±2.61 |
| SISA | | 73.52±0.37 | 0±0 | 50.12±0.02 | | **91.96±0.63** | 0±0 | **61.26±1.42** |
| Unrolling | | 84.26±1.53 | 0±0 | **67.59±2.49** | | 92.48±0.64 | 93.31±2.56 | 57.20±1.64 |
| T-S | | 86.47±0.91 | 6.21±5.07 | 44.95±4.43 | | 92.73±0.64 | 10.29±5.14 | 49.62±0.97 |
| SCRUB | | 32.93±0.84 | 0±0 | 50.59±1.42 | | 20.99±0.31 | **0±0** | 66.13±1.98 |
| **LAF+R** | | **87.15±0.55** | 0.15±0.09 | 58.16±1.08 | | **91.96±0.63** | **0±0** | **61.26±1.42** |
| **LAF** | | 82.38±0.97 | 2.15±1.96 | 50.46±1.96 | | 85.80±1.14 | 0.33±0.51 | 56.33±0.49 |

observed that our proposed LAF-R method achieves the highest performance in 19 evaluations and secures the second-highest performance in 15 out of a total of 44 evaluations. In contrast, the SISA method manages to attain the highest performance in only 13 evaluations and the second-highest in 5 evaluations. Other methods under consideration lag in comparison to LAF-R and SISA. Furthermore, we observe that LAF-R consistently achieves either the best or second-best results in the tasks of data removal and noisy label removal, particularly under the metric of **ASR**. This suggests that the proposed LAF-R stands as a highly reliable unlearning algorithm in countering membership inference attacks. However, a limitation is noted in the class removal task; while LAF-R consistently maintains the highest test data accuracy for the remaining class, it falls short in sufficiently removing the information of the forgetting class. This can be due to the lack of label information, which can compel a shift in the prediction results of the forgetting class to other classes in unlearning (Tarun et al., 2023; Chundawat et al., 2023; Kurmanji et al., 2023).

Regarding the performance of the proposed LAF, it demonstrates comparable results in data removal and noisy label removal tasks, although it exhibits weaker performance in the class removal task. In Table. 1, LAF attains the best and the second best **Train_f** on Fashion and SVHN. It can achieve the best **ASR** on CIFAR10, and the second best **ASR**s on the DIGITS dataset. Moreover, in all evaluations excluding those on the DIGITS dataset, LAF consistently ranks within the top 5 performances. The suboptimal results on the DIGITS dataset can primarily be attributed to the excessive removal of information of the forgetting data, subsequently impacting the performance of the remaining data. In the class removal task, as previously noted, the label-agnostic approach exhibits shortcomings when compared to supervised repairing (LAF-R) and other supervised unlearning methods. In the noisy label removal task, LAF further demonstrates its ability to mitigate the effects of noisy labels and enhance prediction accuracy, securing top-5 rankings in all accuracy evaluations. Furthermore, the

efficacy of the noisy label removal tasks also supports LAF can realize unlearning on low-quality representation extractor maintaining the prediction ability.

Table 3: Comparison results with other state-of-the-art methods in noisy label removal (avg%±std%)

| Method | Data | $Train_r$ | $Train_f$ | Test | ASR | Data | $Train_r$ | $Train_f$ | Test | ASR |
|---|---|---|---|---|---|---|---|---|---|---|
| Retrain | | 99.75±0.12 | 0.17±0.01 | 98.83±0.05 | 39.26±0.01 | | 97.04±0.83 | 2.16±0.06 | 88.15±0.45 | 37.65±1.88 |
| NegGrad | | 98.64±0.22 | 26.26±2.52 | 98.27±0.15 | 30.66±0.93 | | 91.91±1.04 | 2.82±0.43 | 85.96±0.85 | 32.39±1.61 |
| Boundary | | 82.05±9.04 | 7.05±3.03 | 69.85±14.83 | 29.44±1.14 | | 72.82±6.71 | 11.21±1.79 | 54.54±10.37 | 30.58±1.19 |
| SISA | DIGITS | **98.92±4.80** | 1.50±0.04 | **98.80±0.08** | 24.64±0.02 | FASHION | 92.22±5.95 | **1.69±0.06** | **88.90±0.01** | 25.00±0.06 |
| Unrolling | | 67.86±0.26 | 0.43±0.09 | 97.31±0.55 | 31.35±1.10 | | 61.73±1.83 | 3.76±0.83 | 80.02±3.85 | 33.97±1.31 |
| T-S | | 90.71±3.52 | 3.60±1.11 | 83.85±5.69 | 27.05±1.76 | | 85.56±3.13 | 5.64±1.32 | 74.19±5.23 | 28.86±0.78 |
| SCRUB | | 97.27±0.39 | 0.74±0.18 | 96.31±0.63 | 31.48±1.08 | | 87.29±1.35 | 4.29±0.50 | 79.41±2.28 | 33.32±0.26 |
| **LAF+R** | | 98.87±0.19 | **0.23±0.06** | 98.45±0.23 | **35.98±1.41** | | **93.42±0.44** | 2.06±0.20 | 87.71±0.36 | **34.33±0.32** |
| **LAF** | | 96.46±0.67 | 2.70±0.59 | 91.48±1.49 | 18.51±0.57 | | 92.32±0.66 | 4.80±0.71 | 81.21±1.22 | 22.36±0.72 |
| Retrain | | 73.33±0.89 | 7.74±0.23 | 64.74±1.26 | 57.04±0.99 | | 82.46±0.15 | 2.37±0.23 | 93.38±0.35 | 59.55±1.22 |
| NegGrad | | 40.35±5.35 | 8.91±2.09 | 29.97±4.18 | **55.98±0.37** | | 18.48±2.68 | 2.48±1.21 | 17.46±6.08 | **58.25±2.05** |
| Boundary | | 42.69±3.44 | 8.23±1.35 | 33.57±2.04 | 54.81±2.17 | | 44.27±1.43 | 7.86±0.51 | 51.66±1.43 | 58.15±1.12 |
| SISA | CIFAR10 | **69.17±0.11** | **6.75±1.01** | **52.59±0.14** | 28.62±0.02 | SVHN | **80.17±0.13** | 2.45±0.31 | 80.02±0.07 | 44.84±0.04 |
| Unrolling | | 32.81±4.34 | 8.88±2.42 | 32.01±3.87 | 53.86±1.26 | | 29.71±3.00 | 10.52±0.99 | 32.33±5.03 | 53.61±0.58 |
| T-S | | 57.50±2.38 | 10.97±0.83 | 45.92±4.81 | 50.57±1.11 | | 75.45±0.33 | 4.27±0.29 | 83.87±0.56 | 51.16±2.01 |
| SCRUB | | 51.84±1.00 | 10.70±0.41 | 38.06±0.37 | 52.38±1.57 | | 59.89±1.66 | 5.10±0.44 | 66.80±4.96 | 57.22±0.90 |
| **LAF+R** | | 60.49±1.71 | 9.33±0.35 | 51.73±2.27 | 54.49±1.04 | | 79.39±0.27 | 2.85±0.17 | **90.51±0.50** | 55.09±1.64 |
| **LAF** | | 57.44±1.11 | 10.60±0.20 | 47.57±0.63 | 53.18±0.68 | | 77.87±0.35 | 3.59±0.20 | 89.33±0.32 | 51.50±1.17 |

## 4.3 REPRESENTATION SPACE VISUALIZATION

Figure 1(a) and (c) present the visualized distributions of the representations before and after unlearning on the T-shirt class (blue points) and Figure 1(b) shows the representation distributions in the retrained model. In Figure 1(a), the forgetting data of the T-shirt class has a few intersecting distributions with the Shirt class (pink symbols) in the decision boundaries of the two classes. However, in the representation distributions of the retrained model, the cluster of the T-shirt shifts closer to the cluster of the Shirt, resulting in a greater overlap of data points between the T-shirt and Shirt classes. In the representation distributions of the post-unlearning model, which is shown in Figure 1(b), the clusters of the T-shirt consist of two segments. The first segment lies in the decision boundaries of the Dress and Shirt classes because the data of the T-shirt are easily misclassified as these two classes. The second segment near-completely overlaps with the data of the Shirt class, which is consistent with the representation distributions in the retrained model.

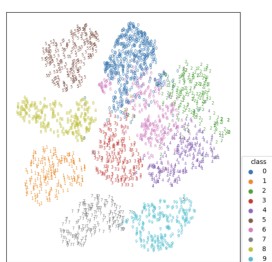
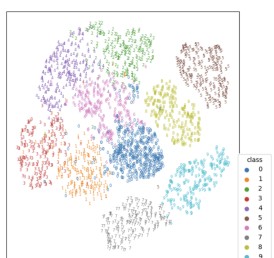
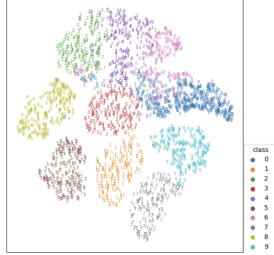

(a) Representation distributions before unlearning

(b) Representation distributions after retraining

(c) Representation distributions after unlearning

Figure 1: Representation distributions in the class removal task on the FASHION dataset. The blue number 0 stand for the forgetting data while the other numbers denotes the remaining data. The colour corresponding to each class is shown in the legends.

## 4.4 ABLATION STUDY

Table 4, 5, and 6 delineate the results of the ablation study focusing on the impacts of losses $L_{UE}$ and $L_{RA}$ across three unlearning tasks. $L_{UE}$ is proposed for extractor unlearning and $L_{UE}$ is formulated for representation alignment to maintain the model's prediction performance. Therefore, it is anticipated that in the absence of $L_{UE}$, the post-unlearning model would exhibit inadequacy in removing forgetting data and without $L_{RA}$, the post-unlearning model will suffer degradation in the prediction performances.

The ablation study results corroborate these expectations. Firstly, in all three tables, the absence of optimization on $L_{RA}$ for alignment with classifiers results in substantial performance degradation in both remaining data accuracy and test accuracy. This is particularly pronounced in class removal tasks. Additionally, in Table 5, the models lacking $L_{UE}$ achieve significantly higher **Test$_f$** on the DIGITS, FASHION, and CIFAR10 datasets, indicating the ability to further remove the information of forgetting data. On the SVHN datasets, although the **Test$_f$** will be lower without $L_{UE}$, there is a notable decline in performance on the remaining data. In Table 4 and 6 where the forgetting data are randomly selected, the forgetting data distribution and the remaining data distribution are similar. Therefore, $L_{UE}$ will have relatively minor influences on the unlearning process and the results without $L_{UE}$ are close to the LAF results in the evaluations on the forgetting data.

Table 4: Ablation study results in data removal. 'None L1' denotes the unlearning without $L_{UE}$ and 'None L2' denotes the unlearning without $L_{RA}$. The following tables take the same notations.

| Method | Data | Train$_r$ | Train$_f$ | Test | ASR | Data | Train$_r$ | Train$_f$ | Test | ASR |
|---|---|---|---|---|---|---|---|---|---|---|
| Retrain | DIGITS | 99.56±0.05 | 98.84±0.10 | 99.04±0.10 | 49.80±0.53 | FASHION | 96.43±0.35 | 92.15±0.41 | 90.23±0.22 | 47.32±0.76 |
| None L1 | | **99.41±0.10** | **99.09±0.45** | **98.81±0.19** | 47.02±1.39 | | 87.30±2.77 | 77.08±7.43 | 81.44±3.45 | 46.05±0.49 |
| None L2 | | 19.02±1.93 | 38.49±15.16 | 22.06±4.38 | 44.62±1.52 | | 44.24±2.47 | 81.68±6.84 | 50.67±3.00 | 40.95±0.07 |
| LAF | | 98.03±0.68 | 97.29±1.43 | 97.30±0.78 | **47.92±0.84** | | **91.54±2.67** | **90.91±7.00** | **87.53±3.26** | **46.89±0.88** |
| Retrain | CIFAR10 | 84.03±0.20 | 78.05±1.34 | 87.20±0.65 | 57.48±0 | SVHN | 83.88±0.23 | 75.16±0.76 | 93.41±0.40 | 58.76±0.48 |
| None L1 | | 78.13±1.28 | 72.96±3.22 | 82.12±2.21 | 56.98±0.79 | | 81.37±0.31 | 71.45±1.26 | 91.41±0.77 | 57.10±0.60 |
| None L2 | | **78.62±0.80** | **80.62±1.59** | **84.67±0.56** | 56.22±0.92 | | 30.52±2.21 | 28.84±5.46 | 40.76±2.48 | 61.81±1.12 |
| LAF | | 78.03±1.55 | 73.30±3.96 | 82.22±2.57 | **57.65±0.70** | | **81.63±0.49** | **76.11±1.49** | **92.32±0.58** | **57.85±0.89** |

Table 5: Ablation study results in class removal.

| Method | Data | Test$_r$ | Test$_f$ | ASR | Data | Test$_r$ | Test$_f$ | ASR |
|---|---|---|---|---|---|---|---|---|
| Retrain | DIGITS | 98.81±0.15 | 0±0 | 26.49±1.41 | FASHION | 92.66±0.29 | 0±0 | 38.24±3.13 |
| None L1 | | **98.88±0.09** | 0.41±0.25 | 24.25±0.70 | | 91.39±0.52 | 8.65±2.10 | 29.48±0.91 |
| None L2 | | 13.97±1.05 | 62.04±37.18 | **26.37±0.92** | | 9.15±1.64 | **1.53±1.47** | 31.28±0.84 |
| LAF | | 98.03±0.68 | **0.26±0.11** | 52.25±2.61 | | **91.54±2.67** | 2.46±1.46 | **31.35±0.71** |
| Retrain | CIFAR10 | 86.01±0.64 | 0±0 | 67.76±1.58 | SVHN | 94.07±0.67 | 0±0 | 59.33±1.31 |
| None L1 | | 7.63±2.22 | 34.93±21.11 | **57.43±3.96** | | 61.13±4.78 | **0.17±0.29** | 61.36±9.89 |
| None L2 | | 33.02±3.61 | 3.60±1.32 | 53.33±4.23 | | 9.54±0.54 | 2.19±3.09 | **60.45±2.57** |
| LAF | | **82.38±0.97** | **2.15±1.96** | 50.46±1.96 | | **85.80±1.14** | 0.33±0.51 | 56.33±0.49 |

Table 6: Ablation study results in noisy label removal.

| Method | Data | Train$_r$ | Train$_f$ | Test | ASR | Data | Train$_r$ | Train$_f$ | Test | ASR |
|---|---|---|---|---|---|---|---|---|---|---|
| Retrain | DIGITS | 99.75±0.12 | 0.17±0.01 | 98.83±0.05 | **39.26±0.01** | FASHION | 97.04±0.83 | 2.16±0.06 | 88.15±0.45 | 37.65±1.88 |
| None L1 | | 90.86±1.00 | 3.74±0.29 | 84.96±1.58 | 29.28±0.55 | | 86.36±1.17 | 5.44±0.72 | 75.46±2.56 | 30.34±0.78 |
| None L2 | | 11.12±2.59 | 11.19±1.39 | 10.85±3.83 | 28.75±1.48 | | 9.47±5.32 | 11.78±1.10 | 8.16±3.40 | **32.41±1.00** |
| LAF | | **96.46±0.67** | **2.70±0.59** | **91.48±1.49** | 18.51±0.57 | | **92.32±0.66** | 4.80±0.71 | **81.21±1.22** | 22.36±0.72 |
| Retrain | CIFAR10 | 73.33±0.89 | 7.74±0.23 | 64.74±1.26 | 57.04±0.99 | SVHN | 82.46±0.15 | 2.37±0.23 | 93.38±0.35 | 59.55±1.22 |
| None L1 | | 57.44±1.10 | 10.61±0.21 | 47.57±0.63 | 53.41±0.52 | | **78.05±0.25** | **3.56±0.27** | 89.10±0.54 | 51.14±0.65 |
| None L2 | | 54.31±2.98 | 10.65±0.24 | 46.86±2.49 | **54.16±1.25** | | 34.63±3.33 | 5.41±1.16 | 43.63±8.15 | **59.24±1.80** |
| LAF | | **57.44±1.11** | **10.60±0.20** | **47.57±0.63** | 53.18±0.68 | | 77.87±0.35 | 3.59±0.20 | **89.33±0.32** | 51.50±1.17 |

## 5  CONCLUSION

In this study, addressing the imperative requirements for unlearning on the label-agnostic datasets, we introduce the Label-Agnostic Forgetting (LAF) framework. This framework is meticulously designed to eliminate the knowledge of the forgetting data distribution, while concurrently maintaining the knowledge of the remaining data at the representational level. Firstly, we employ two VAEs to model the distributions of both training and unlearning data, subsequently introducing a novel extractor unlearning loss to remove the knowledge of the forgetting data. Secondly, we introduce an additional representation alignment loss, intending to align the distributions of the remaining data representations with those preserved in the original model. Finally, if the annotations of any subset of remaining data are available, we proceed to update the entire model through supervised repairing, to further preserve the information of remaining data. The experiment results demonstrate the advantages of the LAF with supervised repairing (LAF+R), in comparison to baseline methodologies. Additionally, the findings also demonstrate the comparable efficacy of LAF without supervisory information, compared to other supervised unlearning approaches. The experiments also shed light on certain limitations of LAF, including the insufficient removal of the forgetting class in the class removal tasks, and the low efficiency compared with other supervised unlearning works. These observed limitations delineate prospective directions for future enhancements and refinements.

ACKNOWLEDGMENTS

The authors thank the NVIDIA Academic Hardware Grant Program for supporting their experiments, the UQ Cyber Security Fund (2021-R3 Weitong) for Shen, the Australian Research Council (DE230101116) for Xu, and the CMS Research Grants (15131570), Adelaide Nottingham Alliance Seed Fund Project (15133470), and ARC (DP240103070) for Chen.

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

# A    SUPPLEMENTARY MATERIAL

## A.1    RELATED WORKS

**Machine Unlearning.**    Machine unlearning requires removing information of forgetting data in the original model while preserving the knowledge contained in the remaining data (Bourtoule et al., 2021; Xu et al., 2024). Currently works on machine unlearning can be summarized into two branches based on the unlearning objectives. The first type is *exact unlearning*, which requires achieving the same model as the train-from-scratch model on remaining data. Exact unlearning is mainly applied to classical machine learning models (Bourtoule et al., 2021; Kim & Woo, 2022). In deep models, the parameters and model structures can be much more complex. Therefore, the exact unlearning is hard to be realized. The current works of exact unlearning on deep models usually take the retraining strategy and focus on improving the algorithm efficiency, for example, the SISA algorithm which retrains the model via a distributed approach on different devices (Bourtoule et al., 2021). The second type is *approximate unlearning*, which requires the unlearned model to get similar performances to the retrained model on both the remaining data and the forgetting data. Approximate unlearning methods are widely applied to deep models (Nguyen et al., 2020; Tarun et al., 2023; Golatkar et al., 2020b; Thudi et al., 2022; Graves et al., 2021; Chen et al., 2023; Kurmanji et al., 2023; Chundawat et al., 2023; Golatkar et al., 2020a; Liu et al., 2021). The cutting-edged works on approximate unlearning includes: (Chundawat et al., 2023) which employs two teacher models that are trained on the remaining and forgetting data to guide the unlearning; (Chen et al., 2023) which explores a new perspective of unlearning by shifting the decision boundary of different classes for unlearning, and (Thudi et al., 2022) which recovers the changes of parameters occurring in the training of data to be forgotten. Compared with the exact unlearning on deep models, approximate unlearning has wider applications.

**Unsupervised Representation Learning.**    Unsupervised representation learning aims to learn the representations of the input data without using labels. The unsupervised representation learning has attracted more attention from researchers. The variational autoencoder (VAE) (Kingma & Welling, 2014) and contrastive learning (van den Oord et al., 2018) are two critical techniques. The VAE can project the input features into low-dimensional Gaussian representations. Currently, the strong ability for representation learning makes the VAE have wide applications on deep learning tasks. For instance, (Liang et al., 2018) explores the application of VAE on representation learning in collaborative filtering while (Kipf & Welling, 2016) applies the VAE on the representation learning of graph data. Contrastive learning reduces the distances of the embeddings of data that share similar characteristics and increases the distances of the embeddings of data that are dissimilar from each other. For instance, (van den Oord et al., 2018) introduces the Noise Contrastive Estimation to differentiate the distance between the similar and dissimilar samples and (Chen et al., 2020) employs cosine similarity during contrastive learning.

## A.2    NOTATIONS

We provide a table of all notations of the main paper in Table 7.

## A.3    IMPLEMENTATION DETAILS

### A.3.1    OVERALL WORKFLOW

Figure. 2 presents the workflow of the whole LAF framework. The LAF first trained two VAEs $h$ and $h_f$ on the representations of training data $\mathbf{X}$ and representations of forgetting data $\mathbf{X}_f$. Then by fixing the parameters of $h$ and $h_f$, Next, to align the representation distribution of $g_U^e$ with the classifier, LAF compares the similarities between the representations of remaining data and forgetting data in the model before and after unlearning and maximizes the representation alignment loss $L_{RA}$. $L_{UE}$ and $L_{RA}$ can be updated alternately. We output the updated model as the final model $g_U^e$.

Subsequently, the LAF framework focuses on aligning the representation distributions between the post-unlearning extractor $g_U^e$ and the classifier $g_D^c$. This is achieved by the representation alignment loss $L_{RA}$, aligning the representations of the remaining data before and after the unlearning process

Table 7: Table of Notations Used in The Main Paper

| Notation | Explanation |
| --- | --- |
| $D$ | Training data |
| $\mathcal{P}$ | Training data distribution |
| $D_r$ | Remaining data |
| $\mathcal{P}_r$ | Training data distribution |
| $D_f$ | Forgetting data |
| $\mathcal{P}_f$ | Training data distribution |
| $x$ | Instance of data |
| $\mathcal{X}$ | Instance space |
| $y$ | Label of data |
| $\mathcal{Y}$ | Label space |
| $g_D$ | Trained deep model |
| $g_D^e$ | Extractor of the trained deep model |
| $g_D^c$ | Classifier of the trained deep model |
| $g_U$ | Post-unlearning deep model |
| $g_U^e$ | Extractor of the post-unlearning deep model |
| $Q(D_r)$ | Distribution that post-unlearning deep model follows on $D_r$ |
| $Q(D_f)$ | Distribution that post-unlearning deep model follows on $D_f$ |
| $\Delta(\cdot, \cdot)$ | Distribution discrepancy |
| $h$ | VAE that learns the distribution of the training data representations |
| $h_f$ | VAE that learns the distribution of the forgetting data representations |
| $\mathcal{N}(0, \mathcal{I})$ | Standard Gaussian distribution |
| $\mu_h, \sigma_h$ | Mean and std estimated by $h$ on its encoding layer for $g_D^e(x), x \in D_r$ |
| $\tilde{\mu}_h, \tilde{\sigma}_h$ | Mean and std estimated by $h$ on its encoding layer for $g_U^e(x), x \in D_r$ |
| $\mu_{h_f}, \sigma_{h_f}$ | Mean and std estimated by $h_f$ on its encoding layer for $g_D^e(x), x \in D_f$ |
| $\tilde{\mu}_{h_f}, \tilde{\sigma}_{h_f}$ | Mean and std estimated by $h_f$ on its encoding layer for $g_U^e(x), x \in D_f$ |

and differentiating the representations of the forgetting data before and after the unlearning. The $L_{UE}$ and $L_{RA}$ losses are updated in an alternating fashion.

The culmination of this process is the final updated model, denoted as $g_U^e$, which effectively embodies the refined balance between learning and forgetting, as dictated by the LAF framework.

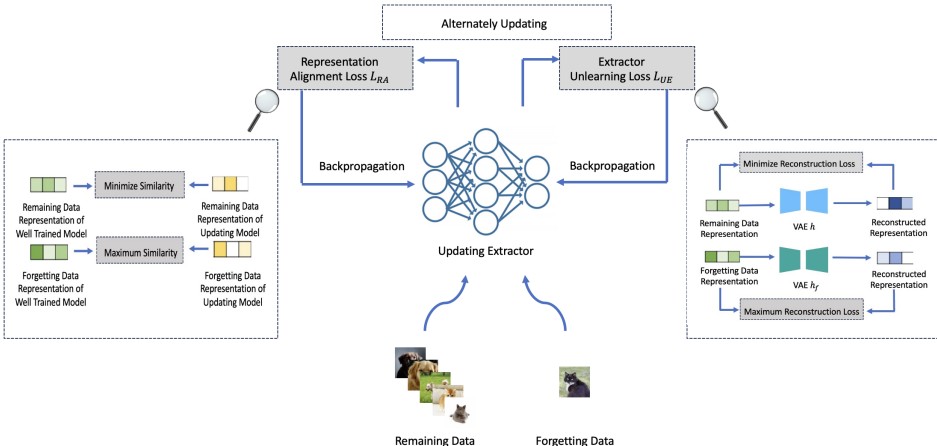

Figure 2: Workflow for LAF consisting of VAE training, extractor unlearning and representation alignment stages.

### A.3.2 ENVIRONMENT

All the experiments are conducted on one server with NVIDIA RTX A6000 GPU (48GB GDDR6 Memory) and 12th Gen Intel(R) Core(TM) i9-12900K (16 cores and 128GB Memory) and two servers with NVIDIA RTX A5000 GPUs (24GB GDDR6 Memory) and 12th Gen Intel Core i7-12700K CPUs (12 cores and 128GB Memory). The code of LAF was implemented in Python 3.9.16 and

Cuda 11.6.1. The main Python packages' versions are the following: Numpy 1.23.5; Pandas 2.0.1; Pytorch 1.13.1; Torchvision 0.14.1. The datasets in experiments: **DIGITS** (LeCun, 1998), **FASHION** (Xiao et al., 2017), **CIFAR10** (Krizhevsky et al., 2009), and **SVHN** dataset (Netzer et al., 2011) are all downloaded from the Torchvision library. Moreover, all the comparison methods provide open resources for their implementation code: **Boundary** [2], **T-S** [3], **SCRUB** [4], **SISA** [5], **Unrolling** [6].

### A.3.3 Initializations

For the experiment models, we choose the **CNN**(LeCun et al., 1995) with two convolutional layers for the two MNIST datasets. The output channels for the two convolutional layers are 16 and 32 respectively. Then the other parts of the CNN consist of three linear layers with the output dimensions 256, 128 and 10. For the two CIFAR datasets, we choose an **18-layer ResNet** (He et al., 2016) with two linear layers with the output dimensions 256, and 10 and the **ResNet** does not contain the pre-trained weights. We construct two VAEs with three and four linear layers in the encoders and decoders. The first type of VAE is used for the two MNIST datasets consisting of three linear layers' encoder with the input dimensions 256, 128, and 32 and a three linear layers' decoder with the input dimensions 8, 32, and 128. The second type of VAE is used for the other two datasets consisting of the same structure encoder as the first one and a three linear layers' decoder with the input dimensions 16, 32, and 128.

All the experiments are based on the original models trained in the four datasets. We train two CNN models on two MNIST datasets for 10 epochs with a learning rate of 1e-3 while we train another two 18-layer ResNet models on two CIFAR datasets for 20 epochs with a learning rate of 5e-5. For the golden standard baselines **Retrain**, we retrain the CNN models on two MNIST datasets for 20 epochs with a learning rate of 1e-3. We retrain the 18-layer ResNet models on two CIFAR datasets for 40 epochs with a learning rate of 5e-5. Then for the other six comparison baselines:**NegGrad**, **Boundary**(Chen et al., 2023), **T-S**(Chundawat et al., 2023), **SCRUB**(Kurmanji et al., 2023), **SISA**(Bourtoule et al., 2021), **Unroll**(Thudi et al., 2022), we keep the hyperparameters of the unlearning process the same as in the original paper and adjust other necessary parameters for the unlearning stage to get as high performances as we can. **NegGrad** adjusts the deep model parameters with positive gradients on remaining data and negative gradients on forgetting data; **Boundary** (Chen et al., 2023) shift the decision boundaries of the forgetting data and remaining data to eliminate the forgetting data information; **SISA** (Bourtoule et al., 2021) proposes to retrain the model using the small data shards from the remaining dataset and ensemble the final results; **Unroll** (Thudi et al., 2022) records gradients when learning the first epoch and adds recorded gradients on weights after the incremental training; **T-S** (Chundawat et al., 2023) proposes to retrain two teacher models on forgetting data and remaining data and adjust the student model through the differences between the output space of the two teacher models; **SCRUB** (Kurmanji et al., 2023) force the model to be consistent with the teacher model trained on remaining data and inconsistent with another teacher model trained on forgetting data.

### A.3.4 Hyperparameters

In all experiments, we configure the batch size to 32. During the training of VAEs, we assign the latent dimensions as 8 for the DIGITS and FASHION datasets and 16 for the CIFAR10 and SVHN datasets. The learning rate for VAE training is established at 1e-3, with the number of training epochs set to 10. For representation alignment, we assign the value of $\tau$ as 2, 20, and 20 for data removal, class removal, and noisy label removal tasks, respectively for CNN. We assign the value of $\tau$ as 20, 20, and 5 for ResNet. Subsequently, in the supervised repairing stage, we designate the repairing epoch as 1, applying a learning rate of 1e-3 for all tasks on the DIGITS and FASHION datasets, and 5e-5 on the CIFAR10 and SVHN datasets.

---

[2]https://www.dropbox.com/s/bwu543qsdy4s32i/Boundary-Unlearning-Code.zip?dl=0

[3]https://github.com/vikram2000b/bad-teaching-unlearning

[4]https://github.com/meghdadk/SCRUB

[5]https://github.com/cleverhans-lab/machine-unlearning

[6]https://github.com/cleverhans-lab/unrolling-sgd

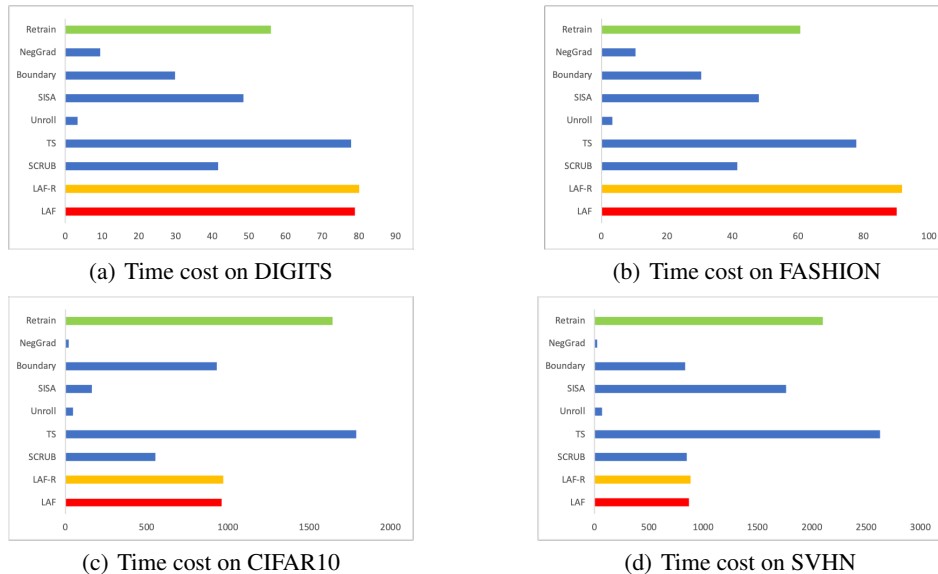

Figure 4: Time cost comparison in the data removal task. The red columns stand for the time costs of the proposed LAF and the orange columns stand for LAF-R. The green columns denote the retraining and the blue columns denote other methods.

## A.4 EFFICIENCY ANALYSIS

### A.4.1 TIME COST ANALYSIS

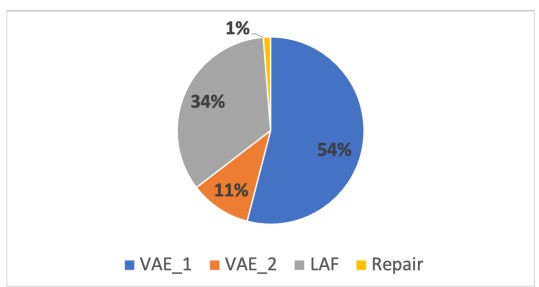

Figure 3: Time cost proportion. VAE_1 stands for the training of $h$ and VAE_2 stands for the training of $h_f$

Figure 4 presents a comparative analysis of the time efficiency of our LAF framework against other methods in data removal tasks. The results indicate that LAF does not hold a distinct advantage in terms of efficiency. Specifically, in experiments conducted on two MNIST datasets, LAF exhibits a slightly higher time cost compared to the seven other evaluated methods. However, in trials involving the CIFAR10 and SVHN datasets, LAF's time consumption is close to the average time cost of other methods and is notably less than that required for retraining and the TS (Teacher-Student) approaches.

This variation in time efficiency primarily stems from the time-intensive process of training the VAEs. As illustrated in Figure 3, the training phase of VAE $h$ accounts for nearly half of the total algorithm runtime, pinpointing a key area for future enhancements. It's important to note, though, that the training of $h$ is conducted on the entire training dataset and is independent of the selection of data to be forgotten. Hence, this training phase can be executed separately from the unlearning process, offering a substantial opportunity to reduce overall time expenditure.

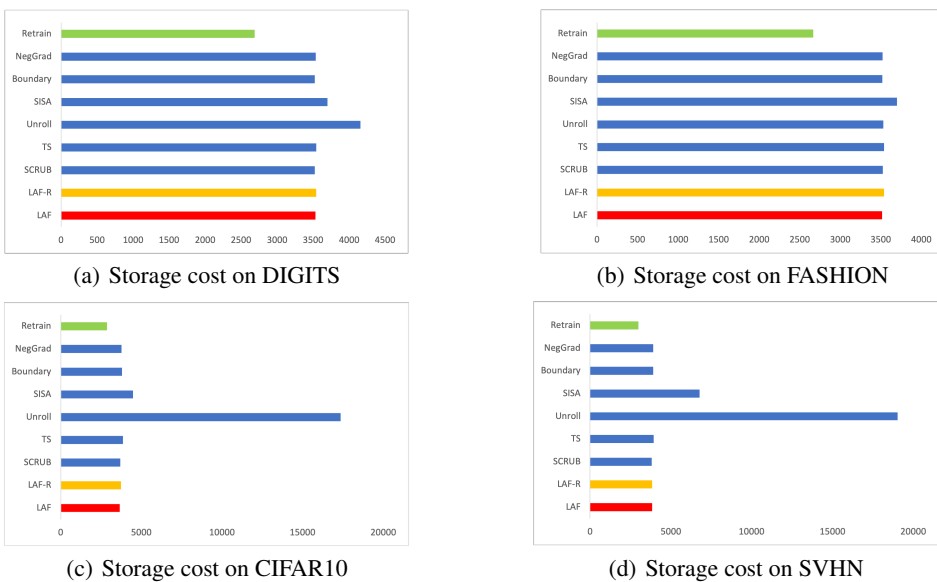

(a) Storage cost on DIGITS          (b) Storage cost on FASHION

(c) Storage cost on CIFAR10          (d) Storage cost on SVHN

Figure 5: Storage workload comparison in the data removal task. The red columns stand for the time costs of the proposed LAF and the orange columns stand for LAF-R. The green columns denote the retraining and the blue columns denote other methods.

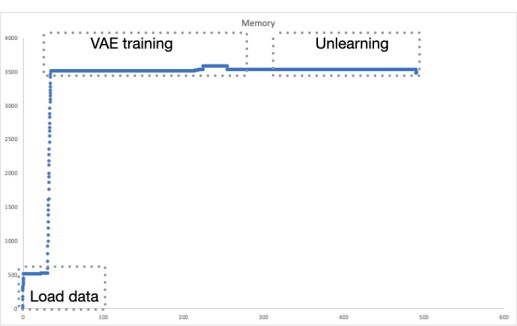

Figure 6: Memory workload changes of LAF during the whole procedure on the random data removal task on DIGITS

### A.4.2 STORAGE WORKLOAD ANALYSIS

Figure 5 provides a comparative overview of the storage workload associated with our LAF framework and other data unlearning methods. The analysis indicates that LAF's storage demands are broadly comparable to those of most other unlearning methods. Notably, the Retrain method exhibits the lowest storage workload, as it does not necessitate any additional memory-intensive components. Conversely, while the Unroll method achieves the lowest time cost, it demands the most storage, particularly in experiments involving ResNet. This increased requirement is due to Unroll's need to store gradients for all parameters across the entire training dataset. Moreover, the SISA approach involves training multiple models concurrently, each mirroring the structure of the original model, thereby escalating the storage requirements. In contrast, our LAF framework avoids the need to store extensive gradients or maintain complex additional models. Although LAF includes the training of two additional VAEs, these are structurally simple, comprising merely five or four linear layers each. For context, the CNN model encompasses 450K parameters, and ResNet-18 contains 11.3M parameters, while the two VAEs collectively have only 150K parameters.

To provide a clearer depiction of the storage workload dynamics within LAF, Figure 6 visualizes the changes in storage requirements throughout the entire LAF process. It reveals that the peak workload

occurs during the VAE training stage, after which the storage demands stabilize during the actual unlearning phase.

## A.5 ADDITIONAL EXPERIMENTS

In this section, we add three parts of additional experiments. A.5.1 is to evaluate the impact of the two approximations in the extractor unlearning process: replacing $D$ by $D_r$ for the training of the first VAE, and dropping of the two KL divergence terms in Eq.8. A.5.2 is to evaluate two different optimization strategies, alternately updating and two-stage updating. A.5.3 is set up to examine the efficacy of the proposed methods on the low-quality representations.

Table 8: Ablation study results in data removal. 'Add KL' adds two KL divergence terms in Eq.8 in the main paper for optimization and $D_r$ denotes training the VAE $h$ using the remaining data. The bold results stand for the best. The following tables take the same notations.

| Method | Data | Train$_r$ | Train$_f$ | Test | ASR | Data | Train$_r$ | Train$_f$ | Test | ASR |
|---|---|---|---|---|---|---|---|---|---|---|
| Retrain | DIGITS | 99.56±0.05 | 98.84±0.10 | 99.04±0.10 | 49.80±0.53 | FASHION | 96.43±0.35 | 92.15±0.41 | 90.23±0.22 | 47.32±0.76 |
| Add KL | | 53.36±3.54 | 85.78±5.14 | 58.90±1.40 | 41.19±0.32 | | 59.97±0.06 | 11.93±2.94 | 48.93±0.23 | 41.57±0.06 |
| $D_r$ | | **99.52±0.01** | **99.43±0.30** | **98.98±0.09** | 56.67±2.61 | | **92.49±0.37** | 90.17±1.57 | **88.22±0.42** | 44.57±0.87 |
| LAF | | 98.03±0.68 | 97.29±1.43 | 97.30±0.78 | **47.92±0.84** | | 91.54±2.67 | **90.91±7.00** | 87.53±3.26 | **46.89±0.88** |
| Retrain | CIFAR10 | 84.03±0.20 | 78.05±1.34 | 87.20±0.65 | 57.48±0 | SVHN | 83.88±0.23 | 75.16±0.76 | 93.41±0.40 | 58.76±0.48 |
| Add KL | | 44.88±32.38 | 40.81±39.45 | 46.33±36.33 | 57.30±5.20 | | **81.92±0.30** | 75.79±0.34 | 91.93±0.32 | 58.09±0.29 |
| $D_r$ | | 77.70±0.67 | **75.59±1.81** | 81.79±0.84 | 55.73±0.73 | | 81.77±0.36 | **75.37±0.82** | 91.88±0.13 | **58.19±0.12** |
| LAF | | **78.03±1.55** | 73.30±3.96 | **82.22±2.57** | **57.65±0.70** | | 81.63±0.49 | 76.11±1.49 | **92.32±0.58** | 57.85±0.89 |

Table 9: Ablation study results in class removal.

| Method | Data | Test$_r$ | Test$_f$ | ASR | Data | Test$_r$ | Test$_f$ | ASR |
|---|---|---|---|---|---|---|---|---|
| Retrain | DIGITS | 98.81±0.15 | 0±0 | 26.49±1.41 | FASHION | 92.66±0.29 | 0±0 | 38.24±3.13 |
| Add KL | | 98.16±0.18 | **0.26±0.05** | 24.83±0.76 | | 88.43±1.24 | **0.75±0.44** | **31.71±0.74** |
| $D_r$ | | **98.18±0.17** | 0.31±0.10 | **24.74±0.80** | | 89.78±0.39 | 2.35±1.04 | 31.45±0.19 |
| LAF | | 98.03±0.68 | 0.26±0.11 | 52.25±2.61 | | **91.54±2.67** | 2.46±1.46 | 31.35±0.71 |
| Retrain | CIFAR10 | 86.01±0.64 | 0±0 | 67.76±1.58 | SVHN | 94.07±0.67 | 0±0 | 59.33±1.31 |
| Add KL | | 47.11±36.02 | **0.10±0.05** | 48.45±1.67 | | 91.14±0.76 | 2.38±2.32 | 54.33±2.47 |
| $D_r$ | | 76.83±0.38 | 2.05±1.65 | 47.14±1.48 | | 90.76±0.14 | 6.31±3.89 | 47.19±3.63 |
| LAF | | **82.38±0.97** | 2.15±1.96 | 50.46±1.96 | | 85.80±1.14 | **0.33±0.51** | **56.33±0.49** |

### A.5.1 FURTHER ABLATION STUDY

Tables 8, 9, and 10 present the findings from our expanded ablation study, focusing on various unlearning tasks. The results highlight that LAF, both in its standard form and with $D_r$ utilized during VAE training, achieves comparable outcomes across most unlearning scenarios. This is particularly evident in tasks involving random data removal. Such consistency validates our approach of substituting $D$ with $D_r$, which offers the advantage of pre-training the VAE, thereby reducing time costs associated with unlearning requests.

Furthermore, upon integrating two KL divergence terms into the optimization process, we observe that performance in class removal and noisy label removal tasks remains similar to both the standard LAF and the LAF with $D_r$ in VAE training. However, a notable difference emerges in random data removal tasks, where we witness a marked decline in performance for the remaining data and test data, along with a greater deviation in attack success rates compared to retrained models. This phenomenon can be attributed to the KL divergence term of the VAE, which, when trained on the entire dataset, acts as a regularization component. This effect makes unlearning more challenging, inadvertently preserving information about the remaining data. It is this observation that led us to exclude these two KL divergence terms from the final extractor unlearning loss formulation.

Table 10: Ablation study results in noisy label removal.

| Method | Data | Train$_r$ | Train$_f$ | Test | ASR | Data | Train$_r$ | Train$_f$ | Test | ASR |
|---|---|---|---|---|---|---|---|---|---|---|
| Retrain | DIGITS | 99.75±0.12 | 0.17±0.01 | 98.83±0.05 | 39.26±0.01 | FASHION | 97.04±0.83 | 2.16±0.06 | 88.15±0.45 | 37.65±1.88 |
| Add KL | | 90.15±1.12 | 3.66±0.17 | 84.40±1.74 | **29.31±0.75** | | 87.82±0.47 | **4.68±0.22** | 78.33±0.75 | 30.19±0.58 |
| $D_r$ | | 90.60±0.59 | 3.53±0.03 | 84.84±1.18 | 28.85±0.66 | | 87.74±0.44 | 4.70±0.19 | 78.27±0.81 | **30.35±0.22** |
| **LAF** | | **96.46±0.67** | **2.70±0.59** | **91.48±1.49** | 18.51±0.57 | | **92.32±0.66** | 4.80±0.71 | **81.21±1.22** | 22.36±0.72 |
| Retrain | CIFAR10 | 73.33±0.89 | 7.74±0.23 | 64.74±1.26 | 57.04±0.99 | SVHN | 82.46±0.15 | 2.37±0.23 | 93.38±0.35 | 59.55±1.22 |
| Add KL | | **77.67±0.90** | **2.80±0.35** | 82.48±0.66 | **51.82±4.76** | | **78.06±2.71** | **3.49±6.43** | 89.11±0.48 | 49.58±0.50 |
| $D_r$ | | 78.31±1.20 | 2.80±0.31 | **82.65±0.56** | 47.18±1.14 | | 78.02±0.08 | 3.53±0.03 | 89.15±0.56 | 50.71±1.16 |
| **LAF** | | 57.44±1.11 | 10.60±0.20 | 47.57±0.63 | 53.18±0.68 | | 77.87±0.35 | 3.59±0.20 | **89.33±0.32** | 51.50±1.17 |

Table 11: Optimizing strategy comparison in data removal.

| Method | Data | Train$_r$ | Train$_f$ | Test | ASR | Data | Train$_r$ | Train$_f$ | Test | ASR |
|---|---|---|---|---|---|---|---|---|---|---|
| Retrain | DIGITS | 99.56±0.05 | 98.84±0.10 | 99.04±0.10 | 49.80±0.53 | FASH | 96.43±0.35 | 92.15±0.41 | 90.23±0.22 | 47.32±0.76 |
| Two Stage | | 88.63±7.06 | 69.22±19.74 | 84.22±9.45 | 44.01±1.29 | | 81.82±0.16 | 71.26±1.37 | **91.28±0.30** | 56.92±0.96 |
| **LAF** | | **98.03±0.68** | **97.29±1.43** | **97.30±0.78** | **47.92±0.84** | | **91.54±2.67** | **90.91±7.00** | 87.53±3.26 | **46.89±0.88** |
| Retrain | CIFAR10 | 84.03±0.20 | 78.05±1.34 | 87.20±0.65 | 57.48±0 | SVHN | 83.88±0.23 | 75.16±0.76 | 93.41±0.40 | 58.76±0.48 |
| Two Stage | | **78.62±0.79** | **80.05±1.11** | **83.51±0.5** | 56.46±0.30 | | **81.82±0.16** | 71.26±1.37 | 91.28±0.30 | 56.92±0.96 |
| **LAF** | | 78.03±1.55 | 73.30±3.96 | 82.22±2.57 | **57.65±0.70** | | 81.63±0.49 | **76.11±1.49** | **92.32±0.58** | 57.85±0.89 |

Table 12: Optimizing strategy comparison in class removal.

| Method | Data | Test$_r$ | Test$_f$ | ASR | Data | Test$_r$ | Test$_f$ | ASR |
|---|---|---|---|---|---|---|---|---|
| Retrain | DIGITS | 98.81±0.15 | 0±0 | 26.49±1.41 | FASH | 92.66±0.29 | 0±0 | 38.24±3.13 |
| Two Stage | | **98.84±0.13** | 1.02±0.31 | **23.59±0.28** | | 91.17±0.17 | 9.05±0.55 | 30.58±0.09 |
| **LAF** | | 98.03±0.68 | **0.26±0.11** | 52.25±2.61 | | **91.54±2.67** | **2.46±1.46** | **31.35±0.71** |
| Retrain | CIFAR10 | 86.01±0.64 | 0±0 | 67.76±1.58 | SVHN | 94.07±0.67 | 0±0 | 59.33±1.31 |
| Two Stage | | 82.27±1.06 | **1.15±0.55** | 46.20±0.72 | | **91.95±0.11** | 2.67±1.98 | 54.78±1.13 |
| **LAF** | | **82.38±0.97** | 2.15±1.96 | **50.46±1.96** | | 85.80±1.14 | **0.33±0.51** | **56.33±0.49** |

Table 13: Optimizing strategy comparison in noisy label removal.

| Method | Data | Train$_r$ | Train$_f$ | Test | ASR | Data | Train$_r$ | Train$_f$ | Test | ASR |
|---|---|---|---|---|---|---|---|---|---|---|
| Retrain | DIGITS | 99.75±0.12 | 0.17±0.01 | 98.83±0.05 | 39.26±0.01 | FASH | 97.04±0.83 | 2.16±0.06 | 88.15±0.45 | 37.65±1.88 |
| Two Stage | | 90.42±0.15 | 3.79±0.16 | 84.12±0.30 | **58.54±0.01** | | 85.52±1.03 | 5.94±0.35 | 73.86±1.63 | **30.08±0.53** |
| **LAF** | | **96.46±0.67** | **2.70±0.59** | **91.48±1.49** | 18.51±0.57 | | **92.32±0.66** | **4.80±0.71** | **81.21±1.22** | 22.36±0.72 |
| Retrain | CIFAR10 | 73.33±0.89 | 7.74±0.23 | 64.74±1.26 | **57.04±0.99** | SVHN | 82.46±0.15 | 2.37±0.23 | 93.38±0.35 | 59.55±1.22 |
| Two Stage | | **80.04±0.33** | **2.67±0.17** | 83.93±0.70 | 57.31±0.22 | | 15.70±0.75 | 12.41±0.18 | 9.65±0.50 | **55.84±0.69** |
| **LAF** | | 57.44±1.11 | 10.60±0.20 | **47.57±0.63** | 53.18±0.68 | | **77.87±0.35** | **3.59±0.20** | **89.33±0.32** | 51.50±1.17 |

### A.5.2 OPTIMIZING STRATEGY

Table 11, 12, 13 presents the results using two different optimizing strategies, alternately updating and two-stage updating. On the DIGITS, FASHION, and SVHN datasets, the alternately updating can reach better forgetting performances and knowledge preservation performances for all three unlearning tasks. In addition, although the two-stage updating can achieve closer results to the retrained models on the preservation of the knowledge from the remaining data, the performances on the forgetting data and the ASR show large differences to the results of alternately updating. Therefore, the experiment results can demonstrate the reasonability and correctness of alternately updating instead of updating in two stages.

### A.5.3 EXPERIMENT ON LOW-QUALITY REPRESENTATIONS

To further examine the efficacy of the proposed LAF, we test LAF with low-quality representations on the different unlearning tasks. Considering that deep models can easily to reach high prediction performances on the two MNIST datasets, we choose the other two datasets: CIFAR10 and SVHN and train two insufficiently trained ResNet-18 models for the experiments. We set the training epochs as 1 and keep the same values of the other hyperparameters as the experiment settings in the main paper. The results are presented in Table 14 and 15.

The sufficiently retrained model and sufficiently trained SISA always reach significantly better performances than all the post-unlearning models because the models provided for unlearning are insufficiently trained. Therefore, the retrained results do not have much reference value in this experiment setting. The results of the original model can prove that all the original models are sufficiently trained and can provide baselines of the performances on the remaining and forgetting data.

Then for the remaining approaches, the results demonstrate that LAF can LAF-R can achieve much better performances than other methods. This can support that LAF can also work on low-quality representation extractors.

Table 14: Comparison results with other state-of-the-art methods in data removal (avg%±std%).

| Method | Data | $Train_r$ | $Train_f$ | Test | ASR | Data | $Train_r$ | $Train_f$ | Test | ASR |
|---|---|---|---|---|---|---|---|---|---|---|
| Retrain | | 84.03±0.20 | 78.05±1.34 | 87.20±0.65 | 57.48±0 | | 83.88±0.23 | 75.16±0.76 | 93.41±0.40 | 58.76±0.48 |
| Original | | 45.59±2.77 | 46.12±2.78 | 48.76±3.95 | - | | 63.21±1.66 | 63.04±1.73 | 72.70±3.18 | - |
| NegGrad | | 20.27±0.93 | 0±0 | 16.20±0.64 | 51.38±0.96 | | 22.42±0.10 | 0±0 | 19.73±0.15 | 60.34±0.06 |
| Boundary | | 21.32±1.34 | 10.40±0.37 | 19.62±2.03 | 54.72±0.81 | | 42.09±1.31 | 12.66±0.19 | 47.21±2.78 | 55.53±1.73 |
| SISA | CIFAR10 | 66.78±0.10 | 53.12±0.74 | 54.30±0.05 | 37.53±0.02 | SVHN | 82.48±0.17 | 67.79±0.34 | 82.57±0.83 | 50.19±0.38 |
| Unrolling | | 27.02±0.16 | 2.28±2.23 | 29.72±0.16 | 57.25±0.87 | | 49.74±1.16 | 14.78±4.44 | 53.43±1.35 | 56.34±0.13 |
| T-S | | 46.48±1.87 | 50.20±4.59 | 50.61±3.14 | 52.98±0.52 | | 64.52±2.20 | 55.16±2.13 | 73.13±4.59 | 55.02±0.25 |
| SCRUB | | 30.00±0.12 | 0±0 | 26.84±0.84 | 53.86±0.55 | | 30.23±0.17 | 0±0 | 27.82±1.04 | 60.30±0.05 |
| **LAF+R** | | 48.11±1.36 | 44.19±1.00 | 52.32±0.50 | 53.43±0.34 | | 68.31±0.55 | 54.77±5.10 | 78.75±0.96 | 55.70±0.57 |
| **LAF** | | 43.55±0.75 | 44.51±0.21 | 46.06±0.87 | 54.95±0.66 | | 63.89±1.20 | 53.94±2.81 | 72.30±3.05 | 54.23±1.41 |

Table 15: Comparison results with other state-of-the-art methods in class removal (avg%±std%)

| Method | Data | $Test_r$ | $Test_f$ | ASR | Data | $Test_r$ | $Test_f$ | ASR |
|---|---|---|---|---|---|---|---|---|
| Retrain | | 86.01±0.64 | 0±0 | 67.76±1.58 | | 94.07±0.67 | 0±0 | 59.33±1.31 |
| Original | | 63.86±9.61 | 47.08±5.27 | - | | 61.27±17.36 | 73.52±3.22 | - |
| NegGrad | | 17.87±0.34 | 0±0 | 46.46±0.43 | | 34.36±0.21 | 0±0 | 64.13±1.74 |
| Boundary | | 35.24±9.22 | 1.52±2.89 | 49.03±1.64 | | 54.36±0.71 | 12.13±1.81 | 61.99±1.20 |
| SISA | CIFAR10 | 99.10±0.03 | 0±0 | 50.12±0.23 | SVHN | 92.14±0.07 | 0±0 | 50.00±0.02 |
| Unrolling | | 42.35±0.67 | 0±0 | 58.55±0.01 | | 60.78±2.68 | 0±0 | 56.85±2.59 |
| T-S | | 48.81±3.05 | 32.30±10.20 | 45.86±2.26 | | 72.10±3.09 | 25.83±14.94 | 57.77±8.49 |
| SCRUB | | 31.45±1.56 | 0±0 | 51.57±0.29 | | 22.53±1.54 | 0±0 | 68.76±2.94 |
| **LAF+R** | | 47.04±0.16 | 0±0 | 47.34±2.35 | | 76.34±0.10 | 0±0 | 56.81±0.65 |
| **LAF** | | 43.07±4.63 | 1.3±0.20 | 43.29±0.34 | | 61.51±4.63 | 0.06±0.06 | 56.67±2.61 |

