# OpenReview forum: "Label-Agnostic Forgetting: A Supervision-Free Unlearning in Deep Models"
_ICLR.cc/2024/Conference — ICLR 2024 poster_

### Official Review · Reviewer_4NxB · 2023-10-28

**Soundness:** 2 fair
**Presentation:** 2 fair
**Contribution:** 2 fair
**Rating:** 5
**Confidence:** 3

**Summary:**

This paper focuses on machine unlearning without the supervision of ground-truth labels during the forgetting process. They introduce a variational approach to approximate the distribution of representations for the remaining data. Leveraging this approximation, they adapt the original model to eliminate information from the forgotten data at the representation level. Experimental results across various forgetting tasks demonstrate the effectiveness of the proposed method.

**Strengths:**

1.	This is the first paper that solves the issue of machine unlearning without labels, which is of great importance.
2.	The experiments are sufficient.

**Weaknesses:**

1.	The technical part is hard to follow. Adding some illustrated figures would be better.
2.	I have a concern about the objective (2). In my opinion, data forgetting is not to maximize the distribution discrepancy to forgotten data. The relationship may be more complex.
3.	The VAE seems redundant. It is utilized to output the features on the given dataset. However, the original model can be directly used instead of training an extra model. In fact, (4) and (6) just mimic the features of the original model. So, the VAE can be viewed as an equivalent version of the original model.
4.	Several typos, e.g., jointly using $g_u^e$ and $g_U^e$. P in (1) should be P_r.

**Questions:**

no

---

> ### Author Response · Authors · 2023-11-17
> **Response to Reviewer 4NxB (Part 1/2)**
>
> We thank you for acknowledging the significance of our research and experiments and for the valuable comments. We answer specific weaknesses in the following.
>
>
> **W1: Adding some illustrated figures for the technical part.**
> ​
> >We thank the reviewer for the suggestion to add an illustrated figure for the technical part. We have added an illustrated figure in Figure 1 in Appendix 3.1. Meanwhile, we also updated the technical part for clarity.
> >#
> >Below we give a brief overview of our methodology part. Our methodology can be divided into optimizing two parts:  extractor unlearning and representation alignment. They are optimized alternatively.
> >#
> > 1. **Extractor Unlearning**:
> >    - **Estimating Distributions**: Estimate distributions of representations for $D_f$ and $D_r$ using the well-trained model.
> >    - **Two Objectives**: Introduce objectives to remove knowledge related to $D_f$ by maximizing the discrepancy between the well-trained model's representation and the unlearned representation on $D_f$, and remaining knowledge related to $D_r$ by minimizing the discrepancy between the well-trained model's representation and the unlearned representation on $D_r$.
> >    - **Optimization**: Merge these two objectives into a single one, and train Variational AutoEncoders (VAEs) to represent these distributions.
> >#
> > 2. **Representation Alignment**:
> >    - **Mitigating Approximation Effects**: Adjust the unlearned representation extractor to align with the original classifier on the classification level.
> >    - **Preserving Predictive Performance**: Introduce a contrastive loss that aligns post-unlearning representations with pre-unlearning ones.
>
> **W2: Concern about the objective (2). Data forgetting is not to maximize the distribution discrepancy to forgotten data. The relationship may be more complex.**
> ​
> >We agree with the reviewer that "Data forgetting is not to maximize the distribution discrepancy to forgotten data. The relationship may be more complex". In light of this, our approach to unlearning aims to condition the model to behave as though it was never trained on the forgetting data. To achieve this, we have established two constraints within our unlearning objective.
> >#
> >The first is to minimize the distribution discrepancy relative to the remaining data, as delineated in Eq.1. This step is pivotal for preserving the maximal amount of knowledge from the remaining data's distribution. The second constraint, outlined in Eq.2, is to maximize the distribution discrepancy concerning the forgetting data. This approach is integral to effectively diminishing the influence of the forgetting data distribution on the model. These two complementary objectives are designed to navigate the complexities of data forgetting, ensuring a more thorough and nuanced unlearning process.

---

> > ### Comment · Reviewer_4NxB · 2023-11-20
> > **Response to W2**
> >
> > The author claims that "The first is to minimize the distribution discrepancy relative to the remaining data, as delineated in Eq.1. This step is pivotal for preserving the maximal amount of knowledge from the remaining data's distribution. The second constraint, outlined in Eq.2, is to maximize the distribution discrepancy concerning the forgetting data. "
> >
> > 1. Why does the first constraint represent the distribution of the remaining data? The representation distribution of the remaining data over the model trained with full data is not equivalent to that of the remaining data over the model trained with only the remaining data.
> >
> > 2. Why does the second constraint represent the distribution of the forgetting data? The representation distribution of the forgetting data over the model trained with full data is not equivalent to that of the forgetting data over the model trained with only the remaining data.
> >
> > Since the two constraints cannot represent their representation distribution separately, it may not be reasonable that the joint constraint can represent the real distribution.

---

> ### Author Response · Authors · 2023-11-17
> **Response to Reviewer 4NxB (Part 2/2)**
>
> **W3: The VAE seems redundant because it is utilized to output the features on the given dataset. Thus, the VAE can be viewed as an equivalent version of the original model.**
>
> >Thank you for your insightful comment. We understand the concern is about the potential redundancy of VAEs; however, the role of VAEs in our framework is significantly different from simply utilizing the features extracted from the original model.
> >#
> >Firstly, VAEs are probabilistic models that offer a distinct advantage in our context. Unlike the deterministic feature extraction of the original model, VAEs are designed to learn the underlying distribution of the extracted features. By training a VAE on these features, we are not just capturing the data points but also modeling the probability distribution that these points follow. This probabilistic representation is pivotal as it allows us to address the unlearning challenge from a distribution perspective, which is not feasible with the deterministic features alone.
> >#
> >Secondly, the unlearning process we have designed plays a crucial role in differentiating our approach. This process involves carefully designed objectives to maximize the discrepancy in representations of the forgetting data and minimize it for the remaining data. Such a targeted approach in the unlearning process, leveraging the distributional capabilities of VAEs, is fundamentally different from and more effective than a straightforward application of the original model's extracted features. It enables a more effective modification of the model in response to the unlearning requirement, ensuring that the forgetting data's influence is minimized while preserving the integrity of the remaining data.
> >#
> >In summary, the use of VAEs in our framework is justified by their ability to model data distributions probabilistically and our specific design of the unlearning process that exploits these probabilistic representations to achieve effective unlearning. This approach goes beyond the capabilities of merely using the original model's extracted features, providing a more sophisticated solution to the unlearning problem.
>
> **W4: Several typos, e.g., jointly using $g_u$ and $g_U$. $P$ in (1) should be $P_r$.**
> ​
> >We appreciate your attention to detail in identifying these typographical errors. We have corrected the instances where $g_u$ was incorrectly used instead of $g_U$, and have also amended $P$ to $P_r$ in Eq.1. We have conducted a comprehensive review of the entire manuscript to ensure that similar errors are rectified and the accuracy of our paper is maintained. Thank you for your valuable feedback.

---

> > ### Author Response · Authors · 2023-11-20
> > **Looking forward to your further feedback**
> >
> > Dear Reviewer 4NxB, thank you again for your thoughtful commentary. Following your suggestions, we included the illustrated figure in the supplementary and addressed other points in our previous response. We would love to hear your thoughts on our response. Please let us know if there is anything else we can do to address your comments.

---

> > ### Comment · Reviewer_4NxB · 2023-11-20
> > **Response to W3**
> >
> > The VAE seems like it uses a generator to provide representations that are similar to the real representations from the full model. The response claims that the generated data is better than the real data due to its probabilistic property. It may not be reasonable. It is obviously better to directly train the model from a real dataset than indirectly train it from the generated dataset by a generator that is trained from the real dataset.

---

> > > ### Author Response · Authors · 2023-11-20
> > > **Response to Feedback of W3 Answer**
> > >
> > > In addition to our previous explanation regarding the use of a model trained on all data, we offer a further response to your question specifically about employing a VAE as a generator within our unlearning framework.
> > >
> > > The primary goal of unlearning is not to train a generator but to adjust the classification model. Instead of generating new samples, we recognize the power of VAE in approximating the real data distribution and want to it to aid the unlearning. In this direction, strategies are needed to connect any trained generator or VAE to a classification model. This paper provides one strategy to connect such that the effectiveness of unlearning can be achieved in this label-agnostic setting.
> > > Specifically, the VAE acts as a tool for adjusting the classification model to achieve unlearning. It is utilized to compute one component of the optimization loss in conjunction with the corresponding data subsets, and plays a pivotal role in our alternate optimization process, guiding the modification of the classification model. The employment of the VAE in this manner is key to our methodology. It is not merely about generating data but about providing a new pathway for the classification model to adapt and unlearn effectively without fully supervised information.

---

> > > > ### Comment · Reviewer_4NxB · 2023-11-21
> > > > **Could you please establish a theory to show the effectiveness of VAE?**
> > > >
> > > > Now, our statements cannot convince one another. The main reason is that there are various intuitive explanations for the proposed method. Theoretical analysis will be more convincing. I would like to see whether VAE can provide some theoretical benefits.

---

> ### Author Response · Authors · 2023-11-20
> **Response to Feedback of W2 Answer**
>
> Thank you for your question concerning the rationale behind using a classification model trained on the entire dataset to train the Variational Autoencoder (VAE) for subgroup data representation, specifically for either the remaining or forgetting data subsets. We understand your point about the ideal scenario where a VAE should be trained directly on the specific data to capture its distribution accurately. Nevertheless, our proposed methodology is based on the following considerations:
>
> - **Limited Data for Forgetting Subset**: Often, the amount of forgetting data is relatively limited, posing challenges in training a VAE from scratch solely on this subset. In contrast, a well-trained classification model, having been trained on a more comprehensive dataset, already encapsulates a substantial understanding of the overall data characteristics. We argue that the output from such a well-trained model, encompassing both forgetting and remaining data, is likely to be more representative than a model trained only on limited data. This premise, assuming the model's comprehensive training and data representation, forms a foundational element of our approach.
>
> - **Preliminary Role of VAE**: Recognizing that the initial training of the VAE may not yield completely accurate representations, it's crucial to note that the VAE primarily serves as a starting point. While not the final representation, the VAE is pivotal in guiding the model's adaptation process, in conjunction with specific data subsets (whether forgetting or remaining data). Our main objective focuses on refining the classification model, where the VAE's precision is less critical but still plays an essential role in facilitating this process. The VAE supports the classification model through subsequent optimization and adjustments, specifically tailored to the respective data subsets. This iterative process enhances the model's capability for more accurate representation learning, which is vital for achieving our unlearning objectives.
>
> - **Efficiency Consideration**: Our approach also considers the efficiency of the unlearning process. Similar to the avoidance of retraining in unlearning tasks due to efficiency and data access constraints, our method circumvents the need for exhaustive VAE retraining from scratch. By leveraging the well-trained classification model as a base, we hasten the VAE's training process, aligning our methodology with practical application requirements.
>
> - **Empirical Validation**: To substantiate our approach, we conducted empirical studies comparing the performance of using the well-trained model's output as an approximation for representations trained using only the remaining data. The results reveal that while training exclusively on the remaining data shows a marginal performance enhancement, the outcomes are closely aligned with those achieved using our proposed method (LAF). This evidence validates our approach, demonstrating its efficacy in approximating the real distribution of data subsets.
>
> | **Data** | **Method**        | **Train_r**        | **Train_f**        | **Test**          | **ASR**          |
> |-------------------|----------|--------------------|--------------------|--------------------|------------------|
> | FASHION  | Retrain           | 96.43±0.35         | 92.15±0.41         | 90.23±0.22         | 47.32±0.76       |
> | FASHION  | Remaining Data    | **92.49±0.37**     | 90.17±1.57         | **88.22±0.42**     | 44.57±0.87       |
> | FASHION  | **LAF**           | 91.54±2.67         | **90.91±7.00**     | 87.53±3.26         | **46.89±0.88**   |
> | CIFAR10 | Retrain            | 84.03±0.20         | 78.05±1.34         | 87.20±0.65         | 57.48±0          |
> | CIFAR10 | Remaining Data    | 77.70±0.67         | **75.59±1.81**     | 81.79±0.84         | 55.73±0.73       |
> | CIFAR10 | **LAF**            | **78.03±1.55**     | 73.30±3.96         | **82.22±2.57**     | **57.65±0.70**   |

---

> > ### Comment · Reviewer_4NxB · 2023-11-21
> > **Could you please directly provide answers to my questions?**
> >
> > My two questions are about the rationale of Eq.1. Your response does not correspond to my qeustions.

---

> ### Author Response · Authors · 2023-11-22
> **Response to Reviewer 4NxB**
>
> **Could you please directly provide answers to my questions?**
>
>
> Thank you for pointing out the need for further clarification. We apologize for any misunderstanding regarding your previous questions.
> For ease of answering the question, let us revisit Equation 1, which we rewrite here after correcting any typos:
>
> $$
> \min_\theta \Delta(Q(D\_r),\mathcal{P}\_r), \text{ where } x \sim \mathcal{P}\_r, g\_{U}^{e}(x) \sim Q(D_r),
> $$
>
> where $\Delta(\cdot,\cdot)$ represents the discrepancy between two distributions, $D_r$ is the remaining data, $\mathcal{P}_r$ is the distribution of remaining data,  and $g_U^e(\cdot)$ is the post-unlearning extractor.
>
> The rationale behind Equation 1 is that it ensures the unlearned model preserves the distribution of the remaining data at the representation level. This is one of the key objectives of unlearning, as it aims to maintain the model’s performance on the remaining data.
>
> In other words, Equation 1 represents an optimization problem that adjusts the model’s parameters. The goal of this optimization is to ensure that the model’s performance remains consistent on the remaining data, even after certain data has been ‘unlearned’ or removed. This approach allows the model to selectively forget specific information while continuing to work effectively with the data that is retained. We hope this explanation provides a clearer understanding of the rationale behind Equation 1.
>
> **Could you please establish a theory to show the effectiveness of VAE?**
>
> Thank you for your insightful comments. We agree with you that a theoretical explanation of why VAEs can aid in our unlearning problem.
>
> While it is true that the theoretical study on VAEs regarding how well they can approximate a data distribution or reconstruct data is still an under-explored area, we believe that the empirical performance of VAEs in approximating data distributions is highly effective and well-recognized. This is the basis for our proposed method.
>
> In contrast to Equation 1, Equation 2, which is also rewritten here for ease of answering this question,
>
> $$
> \max\_\theta \Delta(Q(D\_f),\mathcal{P}\_f), \text{ where } x\sim \mathcal{P}\_f, g\_{U}^{e}(x)\sim Q(D\_f),
> $$
>
> is designed to maximize the discrepancy between the distribution of the forgetting data and the learned representation of the forgetting data by the model. This is achieved by adjusting the parameters of the model in such a way that the learned representation of the forgetting data becomes as dissimilar as possible from the original distribution of the forgetting data.
>
> When we apply the unlearning process, we want to alter this learned representation for the forgetting data, making it dissimilar from the original distribution of the forgetting data. This is where Equation 2 comes into play. By maximizing the discrepancy, we are effectively ‘pushing away’ the forgetting data from its original position in the latent space, causing the model to ‘forget’ this data.
>
> We acknowledge that we cannot provide a rigorous bound on how VAEs can help in this situation. However, our method has been empirically shown to perform well on various unlearning tasks. We hope this explanation provides a clearer understanding of our approach.
>
> If any aspects of our approach seem counterintuitive, we kindly request that you point them out explicitly. This will allow us to address your concerns more effectively and improve the clarity of our work.

---

> > ### Comment · Reviewer_4NxB · 2023-11-23
> > **Final response to authors**
> >
> > Thanks for your response again.  I'm sorry for taking up so much of your time to respond. Yet, I'm afraid I have to disagree with the intuitive explanation of the motivation. I understand the meaning of Equation 1 but I do not agree with the rationale behind it. The real relationship between the representation distribution of remaining data over the forgotten model and of that over the model trained from scratch using the remaining data may not be the minimized discrepancy defined in this equation 1. And a similar case in the forgetting data. It is necessary to establish theories.

---

> > > ### Author Response · Authors · 2023-11-23
> > > **Appreciate the opportunity to discuss with you**
> > >
> > > Dear Reviewer,
> > >
> > > Oh, you do not need to say sorry. We really appreciate your time and effort you have invested in reviewing our paper and giving us feedback. We also appreciate the opportunity you have provided for us to clarify and enhance our work.
> > >
> > > We agree with your statement that Eq 1 is not about "The real relationship between the representation distribution of remaining data over the forgotten model and of that over the model trained from scratch using the remaining data may not be the minimized discrepancy", and we want to seize the opportunity to further clarify our intention in Eq 1. To simply our further statement and ease understanding, we define the following three notations
> > >
> > > $\mathbf{d_u}$: The representation distribution of remaining data over the forgotten model.\
> > > $\mathbf{d}$: The representation distribution over the model trained from scratch using all data.\
> > > $\mathbf{d_{re}}$: The representation distribution over the model trained from scratch using the remaining data.
> > >
> > > Your interpretation suggests that Equation 1 addresses the relationship between $\mathbf{d_u}$ and $\mathbf{d_{re}}$. However, $\mathbf{d_{re}}$ cannot be accessible for unlearning tasks. Instead, the actual intention of Equation 1 is to manage the discrepancy between $\mathbf{d_u}$ and remaining data distribution estimated from $\mathbf{d}$. The primary goal here is not to compare the forgotten model with the model retrained from scratch ($\mathbf{d_{re}}$) but to adjust the pre-unlearning model ($\mathbf{d}$) towards a hypothetical post-unlearning state ($\mathbf{d_u}$).
> > >
> > > Since the retrained model is inaccessible, such adjustment on the pre-unlearning model is crucial for the purpose of maintaining the integrity and performance of the model while adhering to unlearning requirements which can be realized by Equation 2. In addition, some recent works on deep model unlearning all choose a similar strategy to ours to adjust the pre-unlearning model without knowing the exact optimal status $\mathbf{d_{re}}$ [1,2,3].
> > >
> > > We also acknowledge your emphasis on the importance of theoretical foundations in our research. Despite our investigations into this area, we have found that existing theories on approximate unlearning primarily revolve around gradient ascending methods [4,5] and do not readily apply to our framework. Additionally, while VAE is a well-recognized method, it too lacks rigorous theoretical bounds. These challenges notwithstanding, we are motivated by your suggestion to establish theoretical findings. We are keen to explore this direction further, as it could shed light on why our method demonstrates superior empirical performance.
> > >
> > > It's a pleasure to discuss the issues about machine unlearning objectives with you since currently there have been no uniform objectives for deep model unlearning and the publications reframe the objectives based on their own methods. Your insights have been immensely helpful in refining our approach, and we are grateful for the opportunity to discuss and clarify these important aspects of our work. Regardless of the final outcome of this review process, we wish to express our sincere thanks for your guidance and support before the discussion closes.
> > >
> > > Regards,
> > > Submission7132 Authors
> > >
> > >
> > > [1] Chundawat, et al. Can bad teaching induce forgetting? Unlearning in deep networks using an incompetent teacher. AAAI 2023.
> > >
> > > [2] Kurmanji, et al. Towards Unbounded Machine Unlearning. NeurIPS 2023.
> > >
> > > [3] Wang, et al. KGA: A General Machine Unlearning Framework Based on Knowledge Gap Alignment. ACL 2023.
> > >
> > > [4] Sekhari, et al. Remember what you want to forget: Algorithms for machine unlearning. NeurIPS 2021.
> > >
> > > [5] Neel, et al. Descent-to-delete: Gradient-based methods for machine unlearning. ALT 2021.

---

### Official Review · Reviewer_iDSm · 2023-10-29

**Soundness:** 1 poor
**Presentation:** 2 fair
**Contribution:** 2 fair
**Rating:** 3
**Confidence:** 4

**Summary:**

In the context of growing data privacy concerns, various methods for machine unlearning have arisen. However, these methods usually require constant supervision, which can be impractical due to the high cost of labeling real-world datasets. To overcome this challenge, this propose a supervision-free unlearning approach that doesn't rely on labels during the unlearning process. To achieve this, a variational method to approximate the representation distribution of the remaining data is done. Using this approximation, the modified model is able to remove information from the forgotten data at the representation level. To mitigate the lack of supervision, which affects alignment with ground truth, a contrastive loss is to ensure the matching of representations between the remaining data and the original model, thus maintaining predictive performance.

**Strengths:**

1. Quite a relevant problem in real-world applications.
2. Does not require information about the labels.
3. Experiments are extensive. The method shows good performance in the absence of data labels.

**Weaknesses:**

1. No proper explanation in the optimization process. Why some terms are dropped and even if they are considered what could have happened?
2. I think there is a typo in equation 1. The distribution should be P_r instead of P. Similarly for D. Please change the typo. If not please provide an explanation. A lot of notational errors are there such as from argmin equation.8 suddenly becomes argmax in equation.9.
3. There is also a zero-shot method of unlearning which does not assume access to the data at all. This raises questions about the experiment settings. Here you need to have access to the retain and forget set to model with VAE. Is not this assumption too much as in many real-world applications the training dataset remains hidden? So in zero-shot settings, as no data is needed unlearning methods don't assume access to the labels.

**Questions:**

1. In equation 4 the term that is minimized how does it fit a VAE framework. In VAE we maximize the ELBO. Now If we take the KL term in Eq.-4 to be the Corresponding KL term in ELBO how does the first term correspond to the first term in ELBO? A detailed explanation will be better because in VAE we want to maximize the log-likelihood of data and then formulate an ELBO term. In this case, a detailed derivation with proper assumption will help to see how it is a VAE. What do we want to maximize here?
2. If my understanding is correct you drop the terms of KL in the final minimization. So it is simply an encoder without the decoder(as it is generally posed in the VAE framework). Is the formulation as VAE necessary?

---

> ### Author Response · Authors · 2023-11-17
> **Response to Reviewer iDSm (Part 1/3)**
>
> We thank you for all the insightful comments and questions. We respond to specific weaknesses and questions in the following.
>
> **Q1: In VAE we maximize the ELBO. Now If we take the KL term in Eq.-4 to be the Corresponding KL term in ELBO how does the first term correspond to the first term in ELBO? A detailed explanation will be better because in VAE we want to maximize the log-likelihood of data and then formulate an ELBO term. What do you want to maximize here?**
> ​
> > Thank you for your question regarding the formulation of our VAE and its relationship to the ELBO. Your query touches on the core of how we have adapted the VAE framework for our purposes. Let us provide a detailed explanation below. For clarity and ease of understanding, this response employs simplified notations that adhere to the conventional standards of VAE. Please note that these notations may differ from those used in our paper.
> >
> >    In the standard VAE framework, the objective is to maximize the ELBO, which is formulated as:
> >    $$
> >    ELBO = \mathbf{E}\_{z \sim q\_\phi(z|x)}(\text{log } p\_\theta(x|z)) - KL(q\_\phi(z|x) || p(z)).
> >    $$
> >    In our approach, the L2 term in our equation (Eq. 4) can be seen as analogous to the first term of the ELBO. This term represents the expected log-likelihood of the data, given the latent variables.
> >
> >    The derivation of our approach starts with the assumption that the decoder in a VAE, particularly for real-valued inputs, can be approximated as a Gaussian decoder, where $p_\theta(x|z) \sim \mathcal{N}(\mu_\theta(z), \sigma_\theta(z)^2)$. This assumption is based on [1], which posits that the negative log-likelihood of $p_\theta(x|z)$ in this context can be represented as:
> >    $$
> >    -\log{p_\theta(x|z)} = \frac{1}{2\sigma^2}||x - \hat{x}||^2 + D\ln{\sigma} + C,
> >    $$
> >    where $\hat{x}$ is the output of the decoder, and $D$ and $C$ are constants.
> >
> >    Following the insights from [2,3], the optimization process typically treats $\sigma$ as fixed, leading to the simplification of the log-likelihood term to an L2 loss:
> >    $$
> >    -\log{p_\theta(x|z)} = \frac{1}{2}||x - \hat{x}||^2 + C.
> >    $$
> >    Thus, we choose the L2 loss as the reconstruction loss in the optimization process, aligning with the maximization of $\log{p_\theta(x|z)}$ and, consequently, the ELBO.
> >
> >
> > In addition, we have reformulated the first term in our approach to resemble the log-likelihood form for ease of understanding and to highlight its parallel to the ELBO maximization in VAEs. Thank you again for pointing out this issue.
> >
> > **References**
> >
> > [1] Kingma, et al. Auto-encoding variational bayes. ArXiv 2013.
> >
> > [2] Rybkin, et al. Simple and effective VAE training with calibrated decoders. ICML 2021.
> >
> > [3] Liu, et al. Towards visually explaining variational autoencoders. CVPR 2020.

---

> ### Author Response · Authors · 2023-11-17
> **Response to Reviewer iDSm (Part 2/3)**
>
> **Q2 and W1: - Is the formulation as VAE necessary if dropping the terms of KL in the final minimization? - Why some terms are dropped and even if they are considered what could have happened?**
> >#
> >The VAE plays an integral role in modeling the representation distributions, which is essential for our approach. This necessity is evident even when we opt to drop the KL divergence terms during the unlearning stage. Our method leverages the full potential of VAE, utilizing its complete training loss (i.e., Eq.4 and Eq.6) to accurately capture these distributions. This approach differs fundamentally from using a single encoder, underscoring the indispensability of VAE in our model.
> >#
> >In the optimization phase of VAE, the KL divergence term is crucial for effectively projecting input data into a latent Gaussian variable. Thus, during VAE training, we retain the KL divergence terms to optimize the VAEs with the complete loss function. This step ensures accurate modeling of the data distributions, which is vital for the subsequent unlearning process.
> >#
> >For the extractor unlearning phase, we strategically drop the KL terms. This decision is based on our objective to enhance unlearning performance. Preliminary results on the DIGIT and FASHION datasets demonstrate that omitting these KL terms during unlearning leads to better outcomes. The comparison of performances with and without KL terms, i.e., "LAF" and "Add KL" respectively,  provides empirical evidence supporting our approach.
> >#
> >| **Data** | **Method** | **Train_r**     | **Train_f**    | **Test**        | **ASR**       |
> >|----------|------------|-----------------|----------------|-----------------|---------------|
> >| DIGIT    | Retrain    | 99.56±0.05      | 98.84±0.10     | 99.04±0.10      | 49.80±0.53    |
> >| DIGIT    | Add KL     | 53.36±3.54      | 85.78±5.14     | 58.90±1.40      | 41.19±0.32    |
> >| DIGIT    | **LAF**    | **98.03±0.68**  | **97.29±1.43** | **97.30±0.78**  | **47.92±0.84**|
> >| FASHION  | Retrain    | 96.43±0.35      | 92.15±0.41     | 90.23±0.22      | 47.32±0.76    |
> >| FASHION  | Add KL     | 59.97±0.06      | 11.93±2.94     | 48.93±0.23      | 41.57±0.06    |
> >| FASHION  | **LAF**    | **91.54±2.67**  | **90.91±7.00** | **87.53±3.26**  | **46.89±0.88**|
> >#
> >The bold results are the closest ones to the results of the retrained model. More comprehensive results and discussions are detailed in Tables 2, 3, and 4, Appendix 5.1 of our revised supplementary material.
>
>
> **W1: No proper explanation in the optimization process.**
>
> >The optimization process utilizes two key loss functions: extractor unlearning loss and representation alignment loss. In each iteration of the optimization, we optimize the two losses alternately. Below we give a brief overview of the optimizations of the two losses on the Fashion dataset. The details of other datasets have been provided in Appendix 3.
> >#
> > 1. **Extractor Unlearning**:
> >    - **Two Objectives**: Introduce objectives to remove knowledge related to $D_f$ by maximizing the discrepancy between the well-trained model's representation and the unlearned representation on $D_f$, and remaining knowledge related to $D_r$ by minimizing the discrepancy between the well-trained model's representation and the unlearned representation on $D_r$.
> >    - **Optimization**: We first train two VAEs, $h$ on training data and $h_f$ on forgetting data. We then merge these two objectives into a single one, dropping the KL terms. In each optimization iteration, we sample a batch of remaining data to minimize the  normalized L2 loss corresponding to $h$, and sample another batch of forgetting data to maximize the normalized L2 loss corresponding to $h_f$.
> >    - **Optimziers**: Adam optimizier with learning rate as 5e-5.
> >#
> > 2. **Representation Alignment**:
> >    - **Mitigating Approximation Effects**: Adjust the unlearned representation extractor to align with the original classifier on the classification level.
> >    - **Preserving Predictive Performance**: Introduce a contrastive loss that aligns post-unlearning representations with pre-unlearning ones.
> >    - **Optimization**: In each optimization iteration, we sample a batch of remaining data and another batch of forgetting data to calculate the cosine similarity loss between the representations of the original and updated models. Then the two cosine similarity loss are merged into the representation alignment loss for minimization, where the cosine similarity loss of remaining data will be reduced and the cosine similarity loss of forgetting data will be enlarged.
> >    - **Optimziers**: Adam optimizier with learning rate as 1e-4.

---

> ### Author Response · Authors · 2023-11-17
> **Response to Reviewer iDSm (Part 3/3)**
>
> **W2: Typos in equation 1 and other notational errors in Section 3.1**
> ​
> >We apologize for all the typographical errors and thank the reviewer for pointing out them.
> >#
> >The distribution should be $P_r$ and the condition should be $g_{U}^{e}(x)\sim Q(D_r)$ because the unlearning objective requires preserving the knowledge of remaining data. The argmax in equation.9 should be argmin.
> >#
> >We have thoroughly checked all the notation parts of our paper to ensure the notation correctness of the technical part. We want to thank the reviewer again for bringing this up.
> >
> ​
> **W3: Comparison with the zero-shot method of unlearning which does not assume access to the data.**
> ​
> >We agree with the reviewer regarding the importance and potential benefits of a fully zero-shot unlearning approach. Such a method , which does not rely on access to either forgetting or remaining data, including labels, represents an ideal paradigm in unlearning scenarios. It stands out for its minimal reliance on data, aligning well with the increasing emphasis on data privacy and security.
> >#
> >The field of zero-shot unlearning is in its early stages, primarily focusing on class-unlearning problems. The current methods developed in [1] are limited to unlearning entire classes and cannot extend to more complex scenarios. While these methods demonstrate effective forgetting of data, their performance on retaining the information of remaining data appears constrained. This limitation could be attributed to their restrictive data access. As demonstrated below, on class-unlearning tasks, the results indicate that while forgetting is achieved effectively by the zero-shot method GKT, there is room for improvement in preserving the accuracy of the remaining data.
> >#
> >| **Data**  | **Method** | **Test_r**       | **Test_f**       | **ASR**          |
> >|------------|-----------|------------------|------------------|------------------|
> >| DIGITS    | Retrain    | 98.81±0.15       | 0±0              | 26.49±1.41       |
> >| DIGITS    | GKT        | 93.21±0.21       | 0±0              | 50.02±0.04       |
> >| DIGITS    | **LAF**    | 98.03±0.68       | 0.26±0.11        | 52.25±2.61       |
> >| FASHION   | Retrain    | 92.66±0.29       | 0±0              | 38.24±3.13       |
> >| FASHION   | GKT        | 43.19±11.67      | 0±0              | 47.63±0.02       |
> >| FASHION   | **LAF**    | 91.54±2.67       | 2.46±1.46        | 31.35±0.71       |
> >#
> >Recent research efforts, as seen in [2], have been directed towards reducing dependency on the remaining data. However, these approaches still necessitate specifying the data and labels to be forgotten, to generate appropriate training loss and gradients. This requirement indicates that while progress has been made in reducing data dependency, the field has not yet reached the stage of complete zero-shot unlearning, where no data specification is needed at all. In addition, these methods still require label information, which contrasts with our current work where label information is not a necessity for the unlearning process.
> >#
> > **References**
> >
> >[1] Chundawat, et al. Zero-shot machine unlearning. IEEE Transactions on Information Forensics and Security (2023).
> >
> >[2] Sekhari, et al. Remember what you want to forget: Algorithms for machine unlearning. NIPS 2021

---

> > ### Author Response · Authors · 2023-11-20
> > **Looking forward to your feedback**
> >
> > Dear Reviewer iDSm, thank you for your time and thoughtful review. We believe we have addressed your concerns in our response and we would love to know what you think about our previous response. If you’ve any remaining or further questions for us to address, we’re keen to take the opportunity to do so before the discussion period closes. Thank you!

---

> > > ### Author Response · Authors · 2023-11-22
> > > **Looking forward to your further comments**
> > >
> > > Dear reviewer iDSm, I would like to thank you once again for your valuable time and review comments. We have carefully referred to your comments and revised the paper accordingly. We have rewritten part of Section 3.1 and corrected the typos to improve exposition and readability. We also make efforts to address your questions about the optimization process, differences with zero-shot unlearning, maximization of ELBO in VAE training, and why drop KL terms in our response to ensure comprehensibility and technical accuracy. The latest revised version has been uploaded. Since the discussion period will close in the next 12 hours, we sincerely appreciate the opportunity to receive your latest comments and discuss further.

---

### Official Review · Reviewer_1kCq · 2023-10-31

**Soundness:** 3 good
**Presentation:** 2 fair
**Contribution:** 3 good
**Rating:** 8
**Confidence:** 4

**Summary:**

This paper introduces a unique approach to the unlearning problem, addressing a scenario where labels may be available during training but inaccessible in the unlearning phase. This novel perspective enhances privacy protection during unlearning. The paper proposes a supervision-free unlearning method, utilizing a variational technique to model task-relevant representations' distribution. This enables effective information removal from the model. Additionally, a contrastive loss aids in model restoration, mitigating the influence of forgetting data and ensuring performance on the remaining data. The proposed method is rigorously evaluated across various tasks, demonstrating its efficacy in data forgetting, category forgetting, and denoising for noisy data.

**Strengths:**

This paper introduces a novel perspective on the unlearning problem by addressing a scenario where labels may be available during training but become inaccessible during the unlearning phase. This specific problem formulation is different from traditional approaches, which assume continuous access to labelled data throughout the unlearning process. This unique scenario could be a crucial addition to current unlearning, as it reflects real-world situations where label information may be sensitive, noisy, or entirely unavailable during the unlearning phase. This novel formulation expands the scope of unlearning research and adds a crucial dimension to privacy-preserving machine learning techniques.

The paper also offers a potential impact for forgetting in deep models, emphasizing the importance of forgetting representations rather than just the correspondence between representations and labels, aligning with deep learning's characteristic of shared representation information between forgotten and retained data.

The proposed supervision-free unlearning method, leveraging variational techniques to model task-relevant representations' distribution, is a novel contribution. This approach allows for effective information removal from the model and employs a contrastive loss to ensure model restoration, successfully mitigating the influence of forgetting data and maintaining performance on the remaining data. The strength also lies in its comprehensive experimental validation across various tasks, including data forgetting, category forgetting, and denoising for noisy data. This extensive evaluation demonstrates the efficacy of the proposed method.

Regarding clarity, the paper is generally well-written and structured. The problem motivation and formulation are articulated in a clear and organized manner. There could be some improvement though (see weakness and questions for details)

**Weaknesses:**

The explanation of the contrastive loss component, particularly the utilization of the "sim loss," lacks comprehensive coverage, leaving room for confusion regarding which sim loss should be used and why use it.

While the paper is generally well-structured and articulated, there are instances where certain statements could benefit from further elucidation for improved clarity. Specific details are highlighted in the "Questions" section.

**Questions:**

1.	The paper assumes that training data, forgetting data, and remaining data are sampled from different distributions. Could the authors provide an illustrative example or rationale for this particular setup to justify its validity?

2.	In comparison to existing methods, it would be valuable to have an assessment of the time and storage efficiency of the proposed method. Understanding these efficiency metrics would provide additional context for evaluating the method's practicality.

3.	Could the intuitive interpretation of Equation 2 be expounded upon? Further elaboration on the meaning and implications of this equation would aid in enhancing the comprehension.

4.	The proposed method involves two approximations, one pertaining to Equation 4 where "h()" operates on all training data rather than just D_r, and the other involving Equation 8 which removes the second part. Could the authors elaborate on how these approximations may potentially impact the final output and results of the method?

5.	It's noted that both Equation 4 and Equation 6 are presented in the paper. However, it's not entirely clear how these equations are utilized in the proposed method, or alternatively, why they are not used. The paper primarily relies on Equation 5 and Equation 7, which are eventually combined into Equation 8. Additional clarification on the role and application of Equations 4 and 6 would be beneficial for a comprehensive understanding of the method.

---

> ### Author Response · Authors · 2023-11-17
> **Response to Reviewer 1kCq (Part 1/4)**
>
> Thank you for the time and effort you devoted to reviewing our paper, and your recognition of the novelty and potential impact of our approach to the unlearning problem, especially in scenarios where labels become inaccessible during the unlearning phase. In the following, we will address the concerns raised and provide additional clarifications regarding the Weaknesses.
>
> ​
> **Q1: The paper assumes that training data, forgetting data, and remaining data are sampled from different distributions. Could the authors provide an illustrative example?**
> ​
> > Consider the MNIST dataset, which comprises images of handwritten digits, categorized into 10 classes (0 to 9), with each class containing 60,000 images. Suppose our objective is to unlearn the data associated with the digit '1'.
> > #
> > In this scenario, the 'forgetting data' is the subset of the MNIST dataset where the label is '1'. This represents a specific distribution: the conditional probability distribution of an image given it belongs to class '1'.
> >#
> >The 'remaining data', in contrast, would include all images from the other 9 classes (0 and 2-9). This subset has its distinct distribution, characterized by the conditional probability distribution over these nine classes.
> >#
> >The 'training data distribution', from which our model initially learns, is the aggregate or marginal distribution encompassing all 10 classes. It represents the broader, undifferentiated probability distribution of any image in the dataset, irrespective of its class.
> >#
> >In this example of class removal from the MNIST dataset, the training, forgetting, and remaining data can indeed originate from different distributions, each with its unique statistical properties.
>
> ​
> **Q2: Provide additional context for evaluating the time and storage efficiency of the proposed method.**
> ​
> >#
> >We appreciate the suggestion to explore the time and storage efficiency of our framework.
> >#
> >We have included a detailed analysis in Appendix 4.1 of the revised paper, including both time cost (Figures 3 and 4) and storage (Figures 5 and 6) analyses. In this rebuttal, we summarize our findings below on the SVHN dataset to highlight the time and storage efficiency of our LAF method compared to seven other baseline approaches. These results from the table demonstrate that our LAF method not only achieves a favourable balance in time efficiency for larger datasets and models, as evidenced by its average ranking but also maintains a manageable computational workload.
> >#
> >Despite the introduction of two VAEs into our framework, the total computational demand of LAF is comparable to, or even less than, other methods. This efficiency is largely due to the VAEs' streamlined architecture, which comprises less than 150K parameters in total. Despite incorporating two Variational Autoencoders (VAEs) into our framework, LAF's total computational demand remains on par with, or even lower than, other methodologies. This efficiency is largely due to the VAEs' streamlined architecture, which comprises less than 150K parameters in total.
> >#
> > | Method    | Time (s) | Time Rank | Storage (MB) | Storage Rank | Average Rank |
> > |-----------|----------|-----------|--------------|--------------|--------------|
> > | Retrain   | 2079     | 7         | 2998         | 1            | 4            |
> > | NegGrad   | 21       | 1         | 3903         | 5            | 3            |
> > | Boundary  | 870      | 3         | 3900         | 4            | 3.5          |
> > | SISA      | 1722     | 6         | 6781         | 7            | 6.5          |
> > | Unrolling | 88       | 2         | 19032        | 8            | 5            |
> > | T-S       | 2660     | 8         | 3937         | 6            | 7            |
> > | SCRUB     | 885      | 4         | 3809         | 2            | 3            |
> > | **LAF**   | 905      | 5         | 3848         | 3            | 4            |
> >#
> >The bold results are the closest ones to the results of the retrained model.

---

> > ### Author Response · Authors · 2023-11-17
> > **Response to Reviewer 1kCq (Part 2/4)**
> >
> > **Q3: Could the intuitive interpretation of Equation 2 be expounded upon?**
> > ​
> > > To answer this question, we restate Equation 2 here for clarity and ease of presentation:
> > >
> > > $\max_\theta \Delta(Q(D_f),\mathcal{P}_f), \text{ where } x\sim \mathcal{P}\_f,  g\_{U}^{e}(x) \sim Q(D_f)$.
> > >
> > > Here we list the explanation of its components and implications:
> > >
> > > 1. **Symbols and Terms**:
> > >    - $D_f$: The 'forgetting data'.
> > >    - $Q(D_f)$: The distribution of the representation extracted by the post-unlearning model on the forgetting data.
> > >    - $\mathcal{P}_f$: The original distribution of the forgetting data.
> > >    - $x \sim \mathcal{P}_f$: Indicates that the data point $x$ is sampled from the forgetting data's original distribution.
> > >    - $g_{U}^{e}(x) \sim Q(D_f)$: Suggests that the output of the post-unlearning extractor $g_{U}^{e}$ for a data point $x$ follows the distribution $Q(D_f)$.
> > >
> > > 2. **Objective of the Equation and Its Intuitive Explanation**:
> > >    - The goal expressed in this equation is to maximize the discrepancy ($\Delta$) between $Q(D_f)$ and $\mathcal{P}_f$, to ensure that the distribution of the post-unlearning model's output for the forgetting data is as different as possible from the original distribution of that data.
> > >    - Intuitively, this maximization guides the model to 'forget' or 'distort' the characteristics of the forgetting data. By maximizing the discrepancy, we ensure that the model's understanding of the forgetting data is significantly altered.

---

> ### Author Response · Authors · 2023-11-17
> **Response to Reviewer 1kCq (Part 3/4)**
>
> **Q4: The proposed method involves two approximations. How these approximations may potentially impact the final output and results of the method?**
> ​
> >In our approach, the first approximation strategically replaces the remaining data with the complete training data set. This decision is primarily driven by an efficiency perspective: it allows us to utilize the well-trained existing model directly, avoiding the need for retraining with only the remaining data. This efficiency gain is crucial in practical applications where computational resources and time are significant considerations. The discrepancy, introduced by incorporating the forgetting data into the training set, can be effectively controlled and minimized through our proposed dual-loss optimization strategy.
> >#
> >The second approximation in our methodology involves omitting two KL divergence terms from Equation 8, a decision made to enhance the unlearning performance of our model. This approximation is rooted in our focus on the distribution discrepancies during the extractor unlearning phase, specifically targeting the differences between the inputs and outputs of the Variational Autoencoder (VAE). In the context of our model, the KL divergence terms typically act as penalty terms, enforcing a regularization effect. By eliminating these terms, we aim to reduce the constraints on the model, thereby allowing a more flexible adjustment of the distribution during the unlearning process.
> >#
> >To evaluate the analysis, we also add comparison experiments between our proposed LAF and its two without-approximation versions: "Add-KL" and "Remaining" in the following table on two datasets. The results relative to the Retrain demonstrate that after adding the KL terms, the performance dropped. Using the remaining data only may improve the performance a little bit, but still comparable to LAF. More results and details will be provided in Appendix 5.1, mainly in Tables 2, 3, and 4.
> >#
> >To thoroughly assess our methodology, we conducted comparative experiments between our proposed method, LAF, and its two non-approximated versions: "Add KL" and "Remaining Data." The results on the two datasets are summarized in the table below. When compared to the baseline "Retrain" method, the incorporation of KL terms in the "Add-KL" resulted in a noticeable decrease in performance. On the other hand, using only the remaining data ("Remaining Data") shows a slight improvement in performance; however, it remains closely aligned with the results achieved by LAF. These findings are further elaborated with additional experiments and detailed analysis provided in Appendix 5.1, specifically in Tables 2, 3, and 4.
> >#
> >| **Data** | **Method**        | **Train_r**        | **Train_f**        | **Test**          | **ASR**          |
> >|-------------------|----------|--------------------|--------------------|--------------------|------------------|
> >| FASHION  | Retrain           | 96.43±0.35         | 92.15±0.41         | 90.23±0.22         | 47.32±0.76       |
> >| FASHION  | Add KL           | 59.97±0.06         | 11.93±2.94         | 48.93±0.23         | 41.57±0.06       |
> >| FASHION  | Remaining Data    | **92.49±0.37**     | 90.17±1.57         | **88.22±0.42**     | 44.57±0.87       |
> >| FASHION  | **LAF**           | 91.54±2.67         | **90.91±7.00**     | 87.53±3.26         | **46.89±0.88**   |
> >| CIFAR10 | Retrain            | 84.03±0.20         | 78.05±1.34         | 87.20±0.65         | 57.48±0          |
> >| CIFAR10 | Add KL            | 44.88±32.38        | 40.81±39.45        | 46.33±36.33        | 57.30±5.20       |
> >| CIFAR10 | Remaining Data    | 77.70±0.67         | **75.59±1.81**     | 81.79±0.84         | 55.73±0.73       |
> >| CIFAR10 | **LAF**            | **78.03±1.55**     | 73.30±3.96         | **82.22±2.57**     | **57.65±0.70**   |

---

> ### Author Response · Authors · 2023-11-17
> **Response to Reviewer 1kCq (Part 4/4)**
>
> **Q5: How Equation 4 and Equation 6 are utilized in the proposed method and which equations are eventually combined into Equation 8?**
> > #
> >In our proposed method, the central aim is to balance two key objectives: minimizing the discrepancy between the model's representation distribution and that of the remaining data, and maximizing the discrepancy between the model's representation distribution and that of the forgetting data. Within this framework, Variational Autoencoders (VAEs) play a crucial role in learning these representation distributions. Equations 4 and 6 represent distinct optimization objectives that guide the training of these VAEs. Specifically, Equation 4 is employed to train a VAE on the representations of the training data, with the original model's parameters held constant. Concurrently, Equation 6 is used to train a separate VAE on the representations of the forgetting data.
> >#
> >Upon completing the training of the two VAEs, the representation extractor undergoes an update phase to meet the overarching goals. This phase involves minimizing Equation 5 to align the model's representation distribution closer to that learned by the first VAE (representing the remaining data). Simultaneously, it requires maximizing the discrepancy as dictated by Equation 7, focusing on the representation distribution learned by the second VAE (representing the forgetting data). The culmination of these dual processes is embodied in Equation 8, which serves as the comprehensive objective, amalgamating the individual objectives of Equations 5 and 7.
> >#
> >We recognize the need for further clarification on the utilization of Equations 4, 6, and their integration into Equation 8 in our paper. To this end, we will revise our paper to include a more detailed exposition on these aspects. Thank you for pointing out that this part needs revision.
>
> ​
> **W1: Which sim loss should be used and why use it lacks comprehensive coverage.**
> ​
> >In our proposed method, we utilize cosine similarity loss as the sim loss. We choose this cosine similarity because
> >#
> >- Normalization Advantage: Cosine similarity loss is inherently normalized, preventing exponential expansion during representation alignment.
> >- Robustness in Deep Learning: Empirical evidence and research (as cited in references [1,2]) have demonstrated that cosine similarity loss is more robust and effective in deep representation learning contexts.
> >- High-Dimensional Representation: According to findings in reference [3], normalized representations can be interpreted as points on a high-dimensional spherical surface. The distance between two such embeddings is aptly captured by the cosine of the angle between them, hence the relevance of cosine similarity in our context.
> >#
> >We believe that these factors collectively make cosine similarity loss an optimal choice for our method. However, we are open to suggestions from the reviewer regarding alternative similarity measures.
> >#
> >**References**
> >
> >[1] Chen, et al. A simple framework for contrastive learning of visual representations. ICML 2020.
> >
> >[2] Robinson, et al. Contrastive learning with hard negative samples. arXiv 2020.
> >
> >[3] Wang, et al. Understanding contrastive representation learning through alignment and uniformity on the hypersphere. ICML 2020.

---

> > ### Author Response · Authors · 2023-11-20
> > **Looking forward to your feedback**
> >
> > Dear Reviewer 1kCq, we sincerely appreciate your valuable comments on our work. In our previous response and the updated supplementary, we included the new results to address the points raised in your review. We look forward to hearing your further feedback about our response and if there is anything else we can do to improve the work.

---

> > > ### Comment · Reviewer_1kCq · 2023-11-20
> > >
> > > Thank you for your efforts in addressing the comments. I am pleased to see that the questions have been comprehensively addressed. I am happy to increase my score and see this paper being accepted.
> > >
> > > One minor suggestion I have is to include the specifics regarding the statistical significance test employed for the results of Question 4 in the revised version. This additional information would enhance the rigour of your empirical analysis.

---

> > > > ### Author Response · Authors · 2023-11-20
> > > > **Thank you!**
> > > >
> > > > We very much appreciate your kind support, and thank you again for your time and many helpful comments! We will revise our paper accordingly!

---

### Official Review · Reviewer_CMsQ · 2023-11-02

**Soundness:** 4 excellent
**Presentation:** 3 good
**Contribution:** 4 excellent
**Rating:** 8
**Confidence:** 4

**Summary:**

Machine unlearning has been an emerging field due to increasing focus on data privacy. Typically, existing approaches toward unlearning rely on re-learning without forgotten data in a supervised manner, which is not practical since the huge amount of training costs and the need for labeled data. This paper proposes a novel machine unlearning framework without accessing any labels. They adapt the model at the representation level through approximation so that the learned knowledge about forgotten data can be removed. After the approximation, they propose a proper alignment to match changed representations to their original representations via contrastive learning. Empirically, they demonstrate the effectiveness of their unsupervised framework and outperform other supervised methods, which opens a potential new research direction in machine unlearning.

**Strengths:**

* Originality: This paper proposes a novel approach to unlearn without the need of labels and retraining process. They first capture the distribution of training data and forgotten data then unlearn forgotten data at the representation level. Then, through alignment with contrastive learning, they recover the shift for remaining data back to original model. These two steps, remove then recover, are original and novel, especially compared to other supervised approaches.
* Quality:  Though its low efficiency compared to other supervised methods, the results in the experiments showcase its effectiveness. Besides, they consider the scenarios that when a certain amount of supervised information is available, how helpful they would be to repair unlearning model. These shows its quality and the completeness of the paper.
* Clarity: The presentation of this paper is clear and well-organized.
* Significance: Machine unlearning gains more emphasis due to increasing focus on data privacy. This paper proposes a new approach to solve it effectively. Additionally, it’s important to the field to do machine unlearning without using any labels and the need of retraining.

**Weaknesses:**

* The first weakness of this framework is its efficiency. To capture data distribution, a certain amount of instance $x$ and two distribution modeling are needed. Though the framework doesn't need retraining process, framework efficiency and computational workload are encouraged to study and present in the paper.
* The quality of representation extractor may affect framework performance. More representation extractors are needed to be considered to enhance the soundness of this method.
* The availability to access $x$ in training data under machine unlearning scenarios lacks of description. Is this practical in the real-world scenarios?

**Questions:**

* In the implementation, the authors optimized the extractor unlearning loss and representation alignment loss ***alternately***. Does alternately update parameter works better than two-stage learning (saying optimize the extractor unlearning loss with the full epochs then optimize representation alignment loss)?
* (minor comment) Dot labels in Figure 1 can be replaced with their exact labels for better understanding.

---

> ### Author Response · Authors · 2023-11-17
> **Response to Reviewer CMsQ (Part 1/2)**
>
> Thank you for your supportive and insightful review of our paper. We are grateful for your recognition of our work in the context of label-free unlearning approaches. In the following, we will address the questions raised and provide further clarifications on the identified weaknesses as well.
>
> **Q1: In the implementation, the authors optimized the extractor unlearning loss and representation alignment loss alternately. Does alternately update parameter works better than two-stage learning?**
>
> > Yes, we found that alternating the optimization is more effective than a two-stage optimization. The alternating approach is better at addressing the key challenge in machine unlearning: balancing the need to forget certain data while maintaining performance on the remaining data. In a two-stage process, optimizing extractor unlearning loss for full epochs can significantly impair the model's capacity to retain essential information, making subsequent alignment optimization inefficient or even impractical.
> > #
> > To validate our approach, we conducted experiments comparing our method (LAF) with a two-stage learning process.
> > The results, which we will elaborate on in the revised version of our paper, clearly demonstrate the superiority of the LAF method in approximating the performance of the retrain. In our comparison, the method that more closely aligns with the 'Retrain' baseline is highlighted in bold. Specifically, the LAF method excels in preserving performance on the remaining data, outperforming the Two-stage method in this respect.
> > #
> >|   Data   |   Method  |   Train_r      |   Train_f      |   Test         |   ASR          |
> >|----------|-----------|----------------|----------------|----------------|----------------|
> >| DIGITS   | Retrain   | 99.56±0.05     | 98.84±0.10     | 99.04±0.10     | 49.80±0.53     |
> >| DIGITS   | TwoStage  | 88.63±7.06     | 69.22±19.74    | 84.22±9.45     | 44.01±1.29     |
> >| DIGITS   | **LAF**   | **98.03±0.68** | **97.29±1.43** | **97.30±0.78** | **47.92±0.84** |
> >| FASHION  | Retrain   | 96.43±0.35     | 92.15±0.41     | 90.23±0.22     | 47.32±0.76     |
> >| FASHION  | TwoStage  | 81.82±0.16     | 71.26±1.37     | **91.28±0.30** | 56.92±0.96     |
> >| FASHION  | **LAF**   | **91.54±2.67** | **90.91±7.00** | 87.53±3.26     | **46.89±0.88** |
> >#
> >
> >The bold results are the closest ones to the results of the retrained model.
>
>
> **Q2: Dot labels in Figure 1 can be replaced with their exact labels for better understanding**
>
> > Thank you for your suggestion. We have revised Figure 1 in Section 4.3, replacing the dot labels with exact digits for enhanced clarity.
>
> **W1: Though the framework doesn't need a retraining process, framework efficiency and computational workload are encouraged to study and present in the paper.**
> >#
> > Thank you for your suggestion to include studies on the efficiency and computational workload of our framework.
> > #
> > In the revised paper, we have incorporated detailed results in Appendix 4.1, including time cost analysis (Figures 3 and 4) and workload analysis (Figures 5 and 6). In this rebuttal, we summarize our findings below on the SVHN dataset, showing the time and storage efficiency of our LAF and seven other baselines.  These results from the table demonstrate that our LAF method not only achieves a favorable balance in time efficiency for larger datasets and models, as evidenced by its average ranking but also maintains a manageable computational workload.
> > #
> > Despite the introduction of two VAEs into our framework, the total computational demand of LAF is comparable to, or even less than, other methods. This efficiency is attributed to the VAEs' simplistic structures, which contain less than 150K parameters in total. This is significantly lower compared to the parameter counts in more complex models such as CNNs (450K parameters) and ResNet-18 (11.3M parameters). This efficiency in both time and computational resources underscores LAF's practical applicability and versatility in handling various data-intensive scenarios
> > #
> > | Method    | Time (s) | Time Rank | Storage (MB) | Storage Rank | Average Rank |
> > |-----------|----------|-----------|--------------|--------------|--------------|
> > | Retrain   | 2079     | 7         | 2998         | 1            | 4            |
> > | NegGrad   | 21       | 1         | 3903         | 5            | 3            |
> > | Boundary  | 870      | 3         | 3900         | 4            | 3.5          |
> > | SISA      | 1722     | 6         | 6781         | 7            | 6.5          |
> > | Unrolling | 88       | 2         | 19032        | 8            | 5            |
> > | T-S       | 2660     | 8         | 3937         | 6            | 7            |
> > | SCRUB     | 885      | 4         | 3809         | 2            | 3            |
> > | **LAF**   | 905      | 5         | 3848         | 3            | 4            |

---

> ### Author Response · Authors · 2023-11-17
> **Response to Reviewer CMsQ (Part 2/2)**
>
> **W2: The quality of representation extractor may affect framework performance. More representation extractors are needed to be considered to enhance the soundness of this method.**
> ​
> >#
> > We agree that evaluating the performance with various-quality representation extractors is essential to demonstrate the robustness of our proposal and have conducted experiments in data removal and class removal scenarios. Specifically, we extended our experiments to include models with less effective (low-quality) extractors, i.e. CNN [1] and ResNet [2] trained on noisy labelled data rather than clean data. The preliminary results on the FASHION dataset, as seen below, indicate that our LAF method, particularly the LAF-R variant, performs robustly, often achieving the best outcomes. More comprehensive results will be detailed in Table 3, Section 4.2 of our revised paper.
> > #
> > | **Method**  | **Train_r**       | **Train_f**       | **Test**          | **ASR**          |
> > |----------|-------------------|-------------------|-------------------|------------------|
> > | Retrain     | 97.04±0.83        | 2.16±0.06         | 88.15±0.45        | 37.65±1.88       |
> > | NegGrad     | 91.91±1.04        | 2.82±0.43         | 85.96±0.85        | 32.39±1.61       |
> > | Boundary    | 72.82±6.71        | 11.21±1.79        | 54.54±10.37       | 30.58±1.19       |
> > | SISA        | 92.22±5.95        | **1.69±0.06**     | **88.90±0.01**    | 25.00±0.06       |
> > | Unrolling   | 61.73±1.83        | 3.76±0.83         | 80.02±3.85        | 33.97±1.31       |
> > | T-S         | 85.56±3.13        | 5.64±1.32         | 74.19±5.23        | 28.86±0.78       |
> > | SCRUB       | 87.29±1.35        | 4.29±0.50         | 79.41±2.28        | 33.32±0.26       |
> > | **LAF+R**   | **93.42±0.44**    | 2.06±0.20       | 87.71±0.36        | **34.33±0.32**   |
> > | **LAF**     | 92.32±0.66        | 4.80±0.71         | 81.21±1.22        | 22.36±0.72       |
> >#
> > **References**
> >
> > [1] Yann, et al. Convolutional networks for images, speech, and time series. The handbook of brain theory and neural networks 1995
> >
> > [2] Kaiming, et al. Identity mappings in deep residual networks. In
> ECCV 2016
> ​
>
> **W3: The availability to access $x$ in training data under machine unlearning scenarios lacks description. Is this practical in the real-world scenarios.**
> ​
> > We appreciate the reviewer pointing out the need for clarification on the availability of training data in our framework. Our method indeed necessitates access to a subset of training data subsampled uniformly to conveying information of the training data distribution. This is essential for training the Variational Autoencoder (VAE).
> > #
> >
> > It is noteworthy that in many academic studies, the assumption of having access to training data, or at least a subset of it, is a common approach in the domain of deep model unlearning, as evidenced by several key publications in this field [1,2,3,4]. While there are efforts to reduce dependency on training data [5], this method limited generally limited to scenarios like class removal and are not feasible for tasks such as data removal or noisy label removal. In addition, [6] is a zero-shot method which do not require both training and forgetting data. But it can only be limited to class removal and its performance drops significantly after unlearning.
> > #
> > In practical applications, consider a social media platform that uses a machine learning model to recommend content to users. This model is initially trained on a large dataset comprising user interactions, such as likes, comments, shares, and viewing times. Over time, due to privacy concerns or user requests, the platform may need to unlearn specific data points related to certain users' interactions. For example, if a user requests to remove his records in recent days, his previous records can still be accessed to recommend content to him. In this example, both training data and forgetting data can be accessed, but the platform may not want to retrain the model from scratch using only remaining data.
> >#
> > **Reference**
> >
> > [1] Chundawat, et al. Can bad teaching induce forgetting? Unlearning in deep networks using an incompetent teacher. AAAI 2023.
> >
> > [2] Thudi, et al. Unrolling sgd: Understanding factors influencing machine unlearning. EuroS\&P 2022.
> >
> > [3] Bourtoule, et al. Machine unlearning. SP 2021.
> >
> > [4] Tarun, et al. Fast yet effective machine unlearning. TNNLS 2023.
> >
> > [5] Chen, et al. Boundary Unlearning: Rapid Forgetting of Deep Networks via Shifting the Decision Boundary. CVPR 2023.
> >
> > [6] Chundawat, et al. Zero-shot machine unlearning. IEEE Transactions on Information Forensics and Security (2023).

---

> ### Comment · Reviewer_CMsQ · 2023-11-18
>
> Thanks for the response! The analysis for efficiency and storage looks great and promising! The discussion about accessing training data is encouraged to add into paper. I don't have any further questions/concerns to ask, and I support this paper to be accepted.

---

> > ### Author Response · Authors · 2023-11-20
> > **Thank you!**
> >
> > Thank you for your encouraging support, and thank you again for your valuable time and insightful comments and suggestions!

---

### Author Response · Authors · 2023-11-22
**Changes in revisions**

We thank the reviewers for their precious time and insightful comments that improved the exposition and readability of our paper. We have made the following changes in our revision and the supplementary file accordingly:
1. We rewrite part of Section 3.1 and polish Algorithm 1 to make the technical part easier to follow to address the weakness from Reviewers iDSm and 4NxB.
2. We fix some typos and notation errors in Section 3.1 following the comments of Reviewers iDSm and 4NxB.
3. We add statements for the usage of cosine similarity in Section 3.2 based on the comments of Reviewer 1kCq.
4. We fix the legend in Figure 1 in Section 4.3 following the comments of Reviewer CMsQ.
5. We add a notation table in Table 1 in Appendix 2.1.
7. We add a workflow figure in Figure 2 in Appendix 3.1 following the suggestion of Reviewer 4NxB.
7. We add time and storage efficiency analysis in Appendix 4 to address the weakness pointed out by Reviewers CMsQ and 1kCq.
8. We add three extra experiments and present the results in Appendix 5 as recommended by Reviewers CMsQ and 1kCq.

---

### Meta-Review · Area_Chair_76y7 · 2023-12-06

**Metareview:**

This paper presents a machine unlearning method which is based on eliminating  representation level information from the data to be forgotten. The proposed unlearning method doesn't require labels of the forget or retain set which. This type of "label agnostic" unlearning, to my knowledge, is a novel contribution in the area of machine unlearning.

The reviews for this paper were positive in general with 2 reviewers strongly recommending acceptance and one being borderline but with comments that were adequately addressed in the rebuttal. One of the reviewers voted for rejection but without significant reasons (one of which was regarding the existence of works on zero-shot unlearning which however is an orthogonal direction). These concerns were also well-addressed by the rebuttal.

Based on the reviews, the authors' rebuttal, and my own reading of the work, I recommend acceptance. The problem setting is novel and the proposed solution is sound with good empirical evaluation. The authors are advised to incorporate the suggestions from the reviewers when preparing the final version.

**Justification For Why Not Higher Score:**

The paper is somewhat weak due to the lack of theoretical guarantees which are usually reassuring for unlearning methods. For this reason, I am only recommending accept (poster).

**Justification For Why Not Lower Score:**

The paper has sufficient merits for acceptance as mentioned in the meta-review.

---

### Decision · Program_Chairs · 2024-01-16

Accept (poster)